# ALAS: Multi-Agent LLM Planning System via Validator Isolation and Localized Cascading Repair Protocol

## Abstract

Large language models enable flexible multi-agent planning but struggle with reliability: verification is often circular, state changes are not tracked for repair, and small faults trigger costly global replanning. We present *ALAS*, a multi-agent LLM Planning framework that separates planning from non-circular validation and performs localized repair guided by versioned execution logs. The validator operates independently of the planning LLM with fresh, bounded context, avoiding self-check loops and mid-context attrition. The repair protocol edits only the minimal affected region of the plan while preserving work-in-progress. On urban ride-sharing and job-shop scheduling across five classical benchmarks, the system matches or exceeds state-of-art Single-LLM and Multi-Agent System baselines, which achieves 83.7% success rate, reduces 60% token usage, and 1.82× Faster. A minimal reliability study shows that the validator detects injected structural faults with low overhead, and localized repair contains runtime perturbations with bounded edit radius and reduced makespan degradation versus global recompute. Code and seeds will be released. Results indicate that ALAS the combination of validator isolation and localized repair with execution logs provides measurable efficiency, optimality, and scalability for multi-agent LLM planning.

## 1 Introduction

Large language models (LLMs) have transformed language understanding and generation (Luo et al., 2025; Matarazzo & Torlone, 2025; Minaee et al., 2025). Yet when used for *planning*, standalone LLMs remain brittle: they often produce incomplete or inconsistent action sequences, violate constraints, and struggle to revise partial plans under change. These failures become acute in settings that require long-range consistency, multi-entity coordination, and reactive behavior.

The causes are structural. First, verification is frequently *circular*: the same model (or context) that proposes a plan is asked to approve it (Hong et al., 2024), inviting rubber-stamping. Second, long contexts are prone to information loss and mid-context attrition (Hsieh et al., 2024; Liu et al., 2024; Vaswani et al., 2017; Xiao et al., 2024). Third, maximum-likelihood decoding biases search toward high-probability but not necessarily high-feasibility completions (Chang, 2023; Holtzman et al., 2020; Radford et al., 2019). Finally, without external state, LLMs cannot reliably track commitments, causal dependencies, or temporal constraints, leading to cascading errors across reasoning chains (Chu et al., 2024a; Patel et al., 2024; Xiong et al., 2024).

Dynamic environments amplify these issues. In logistics, event coordination, or industrial operations, late requests, delays, and resource failures require *local* edits to in-flight plans. Global recomputation—whether via classical optimization or re-prompting a single LLM—can be counterproductive: small disturbances trigger large plan changes, harming latency and stability, and often break feasibility along the way.

**Our approach.** We study *efficiency*, *optimality*, and *scalability* in multi-agent LLM planning as a systems property of the end-to-end process. We present Adaptive LLM Agent System (ALAS), a stateful, disruption-aware framework that (i) separates *planning* from *non-circular validation* (validator isolation), (ii) records state transitions in a *versioned execution log* that provides restore points and auditability, and (iii) applies a *localized repair* protocol that edits only the minimal

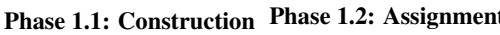

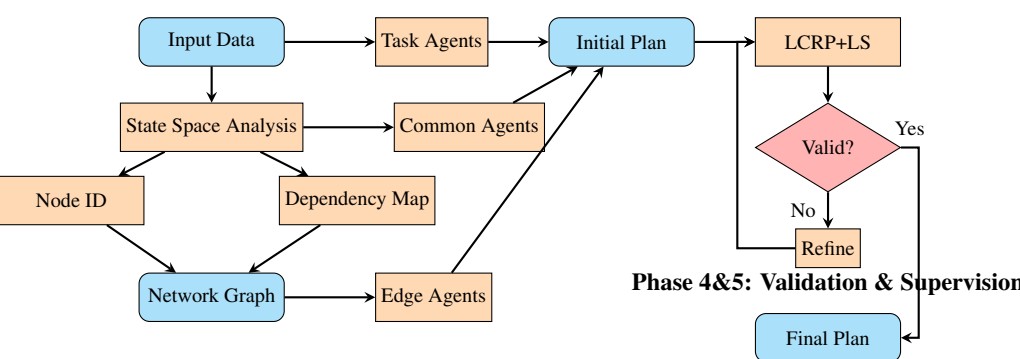

Figure 1: ALAS Planning Layer #1 Architecture. Color Scheme: cyan for input/output/intermediate results, orange for processes, and red for decisions. This figure illustrates how the meta-planner generates a planning workflow template $\mathcal{W}_{\text{template}}$ with phases arranged from left to right.

affected region of the plan while preserving work-in-progress (WIP). By grounding validation in fresh, bounded prompts over the execution log, ALAS avoids self-check loops and mid-context drift; by repairing locally, it contains faults without costly global replanning.

**Agent roles.** Rather than a monolithic planner, ALAS orchestrates lightweight agents specialized to known failure modes:

1. *Validator agents* (isolated from planners) check feasibility against constraints and execution log.
2. *Repair agents* compute change sets under constraints to localize edits when disruptions occur.
3. *Context agents* operate in semantically scoped subcontexts to mitigate long-context attrition.
4. *Monitoring agents* detect anomalies/events and trigger validation or repair.
5. *Execution-log modules* maintain versioned state and restore points for audit and recovery.

**Contributions.**

1. *Validator isolation.* We decouple planning and verification: a dedicated validator operates with fresh, bounded context over the execution log, preventing circular self-approval and mitigating mid-context loss.
2. *Localized repair.* A disruption-aware protocol computes bounded edit sets that preserve WIP and constraints, avoiding brittle global recomputation.
3. *Versioned execution logs.* We record state transitions and restore points for auditable updates and targeted recovery (without relying on monolithic long contexts).
4. *Evidence across domains.* On URS and job-shop scheduling (DMU, TA), our framework matches or exceeds strong single-LLM and classical baselines, with the largest gains under disruption. We further provide *minimal reliability evidence*: a small fault-injection and runtime-perturbation study showing high detection rates for injected structural faults and bounded repair with modest token/latency overhead.

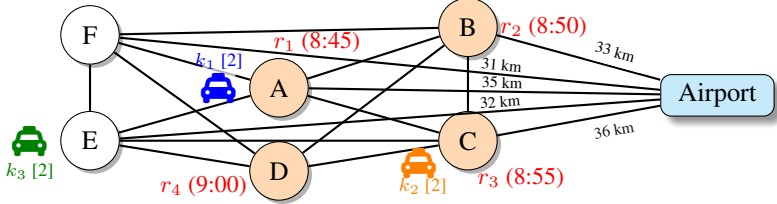

Figure 2: Network $G = (V, E)$ with urban travel times $\tau_{ij} = 10$ minutes and airport routes distance specified on the figure. Static scenarios can be solved by MILP or Column Generation. Dynamic scenarios (e.g., an accident, a cancellation, new passenger requests) must be addressed by ALAS. We adopt *Urban Ride Sharing* (URS) as a running example (Fig. 2): multiple vehicles must deliver passengers under delays and last-minute requests. Unlike TSP Lawler et al. (1985), URS demands

concurrent execution, inter-agent coordination, and reactive edits. Section 4 then evaluates more complex job-shop settings.

We emphasize reliability under disruption, not perfect generation. The results indicate that *non-circular validation plus localized repair over execution logs* yields measurable robustness for multi-agent LLM planning in dynamic environments.

## 2 RELATED WORK

We group related work into: (i) structural limitations of LLMs for planning, (ii) LLM-based multi-agent systems and orchestration, and (iii) LLM planning frameworks and benchmarks.

### 2.1 STRUCTURAL LIMITATIONS OF LLMS

Transformer LLMs (Brown et al., 2020; Vaswani et al., 2017) excel at language tasks but face well-documented challenges in planning: *circular verification* (the proposer approves itself) (Hong et al., 2024), *solution-space bias* from maximum-likelihood decoding (Chang, 2023; Holtzman et al., 2020; Radford et al., 2019), *context degradation* in long inputs (Hsieh et al., 2024; Liu et al., 2024; Xiao et al., 2024), *error propagation* across multi-step chains (Chu et al., 2024a; Patel et al., 2024; Xiong et al., 2024), and the *absence of persistent state*. While self-reflection, CoT variants, and structured validators can help in narrow domains (Madaan & Yazdanbakhsh, 2022; Li et al., 2023b; Jiang et al., 2024; Feng et al., 2023; Gou et al., 2024), open-domain plan validation and reliable revision remain open problems (Chen et al., 2024; Huang et al., 2024). We target these failure modes by (i) separating planning from *non-circular* validation (*validator isolation*), (ii) maintaining a *versioned execution log* that grounds checks and provides restore points, and (iii) using *localized repair* to bound edits during disruption.

### 2.2 LLM MULTI-AGENT SYSTEMS AND ORCHESTRATION

LLM-based multi-agent systems enable role specialization and workflow graphs for tool use and collaboration (e.g., AutoGen, MetaGPT, LangGraph, CAMEL) (Wu et al., 2024b; Hong et al., 2023; LangChain AI, 2024; Li et al., 2023a). Other lines explore graph-structured or programmatic coordination (e.g., GPTSwarm, Flow/AFlow) (Zhang et al., 2024). These advances emphasize *orchestration*, but plan execution is typically best-effort generation without: (1) a *non-circular* validation pathway, (2) *versioned execution logs* with restore points for auditable state, or (3) a *localized repair* protocol that constrains the impact radius of faults. Independent evaluations have also reported mixed gains of MAS over single-agent baselines on broad suites, citing miscoordination and unmet preconditions (Cemri et al., 2025; Trivedi et al., 2024; Qian et al., 2023; Phan et al., 2024).

A complementary systems line studies *durable* saga-style execution and compensation for agent workflows (e.g., SAGALLM (Chang & Geng, 2025)). That work focuses on execution-layer durability and recovery semantics. In contrast, our focus here is *planning reliability*: we separate planning from validation to avoid circular self-approval, ground checks in *versioned logs*, and perform *bounded local repair* instead of global recompute.

### 2.3 LLM PLANNING FRAMEWORKS AND BENCHMARKS

**LLM-based planning.** LLMs have been applied to decomposition and search (e.g., PLASMA, LLM-MCTS) and to multi-agent workflows (AFlow and variants) (Brahman et al., 2024; Zhao et al., 2023; Zhang et al., 2024). Some systems attempt direct LLM solving for scheduling or optimization (Abgaryan et al., 2024; 2025), but classical methods (e.g., SBP, CP/Tabu, metaheuristics) remain stronger on static benchmarks (Adams et al., 1988; Nowicki & Smutnicki, 1996; Aarts & Van Laarhoven, 1989; Bierwirth, 1995; Kirkpatrick et al., 1983). We leverage LLMs for *interpretability and adaptive repair* under disruption, aligning with the "reason+act" paradigm (Yao et al., 2023) while adding execution logs and localized repair that these methods lack.

**Benchmarks.** Common LLM benchmarks (e.g., HotPotQA, ALFWorld, BIG-Bench) stress static reasoning, with recent datasets adding temporal structure (PlanBench, TimeBench, ACPBench) (Yang et al., 2018; Shridhar et al., 2021; Srivastava et al., 2022; Valmeekam et al., 2023; Chu et al., 2024b; Abdin et al., 2024). Disruption-aware evaluation remains limited. We therefore use classical Job-Shop Scheduling—Demirkol-DMU and Taillard (TA) (Demirkol et al., 1998a; Shylo et al., 2018; Xiong et al., 2022)—and introduce runtime perturbations (e.g., machine downtime, operation-delay shocks).

These benchmarks provide formal constraints, known bounds, and natural perturbation models, making them suitable for assessing execution logs, validator isolation, and localized repair.

## 3   ALAS: A FIVE-LAYER ARCHITECTURE

Existing orchestration frameworks rarely guarantee reliable planning under dynamic, stateful conditions. We introduce ALAS, a three-layer architecture that turns a high-level specification into a validator-isolated, locally repairable execution workflow. The layers are: (i) *workflow blueprinting*, (ii) *agent factory*, and (iii) *runtime execution & localized repair*. A *versioned execution log* persists state and provides restore points for validation and repair.

---

**Algorithm 1** Phase 1: Workflow Template Construction by $\mathcal{T}$ (summary). Details in App. B.1.

---

**Require:** Task specification $\mathcal{O}$; constraint set $D$; (optional) disruption model $\Phi$
**Ensure:** Validated template $\mathcal{W}_{\text{template}} = (\mathcal{N}, \mathcal{E}, \mathcal{C}, \mathcal{L})$
 1: Extract abstract roles $\mathcal{R}$ from $\mathcal{O}$
 2: Map roles to nodes $\mathcal{N}$ with profiles $\mathcal{P}_{n_i}$
 3: Derive dependencies $\mathcal{E}$ under constraints $D$; collect invariants $\mathcal{C}$
 4: For each $n_i \in \mathcal{N}$, attach role spec $\alpha_i = \langle \text{cap}_i, \text{ctx}_i, \text{io}_i, \mathcal{L}_i \rangle$
 5: For each $n_i$, attach **repair spec** $\rho_i$ (local edit primitives; scope/bounds)
 6: Define **versioned execution log** schema $\mathcal{L} = \{\mathcal{L}_i\}_{n_i \in \mathcal{N}}$ (events, snapshots, diffs)
 7: Attach **independent validator** $V$ with fresh, bounded prompt scope $\kappa$ (planner $\neq V$)
 8: **while** $\mathcal{W}_{\text{template}}$ fails validation by $V$ **do**
 9:     $V$ checks: (i) structural soundness; (ii) constraint satisfaction vs. $\mathcal{C}$;
            (iii) **repair coverage**: for disruptions in $\Phi$, local neighborhoods and edit bounds exist
10:     Refine nodes, edges, role specs, or repair scopes; update $\mathcal{L}$
11: **return** $\mathcal{W}_{\text{template}} = (\mathcal{N}, \mathcal{E}, \mathcal{C}, \mathcal{L})$

---

### 3.1   LAYER 1: WORKFLOW BLUEPRINTING (LLM TEMPLATE CONSTRUCTION)

Given a planning input $\mathcal{O}$ (goals, resources, constraints, disruption model), ALAS synthesizes a workflow template

$$\mathcal{W}_{\text{template}} = (\mathcal{N}, \mathcal{E}, \mathcal{C}),$$

where nodes $\mathcal{N}$ are abstract *roles* (planner/validator/repair/monitor/domain), edges $\mathcal{E}$ encode data/control dependencies, and $\mathcal{C}$ collects global constraints and invariants. Roles are specified independently of concrete instances (e.g., "pickup at zone $z$" rather than a specific driver), enabling late binding at execution.

In LLM Query Agent – 'generate_schedule()', the process begins with an LLM-based query agent that produces an initial candidate schedule given the problem specification (jobs, machines, constraints). Schedules are either generated fresh or loaded from pre-computed JSON outputs (e.g., 'singleagent_llm_comparison' results). Metadata such as makespan and entry counts are logged at this stage.

**Phase I: Graph sketch.** From $\mathcal{O}$, we draft a directed acyclic graph of roles and dependencies. Roles may remain unresolved (e.g., unassigned vehicle, unbound machine slot).

**Phase II: Role specs.** Each role is annotated with: required capabilities, input/output schemas, a short *context scope* (to bound prompts), and a logging schema for the execution log. Repairable edges (time/order/resource) are marked so that only the minimal neighborhood is considered during disruption.

**Phase III: Agent Factory (Instantiation from Specs)**

The agent factory turns role specs into executable agents and binds unresolved roles to concrete instances when possible. For role $i$ we use a lightweight signature

$$\alpha_i = \langle \text{cap}_i, \ \text{ctx}_i, \ \text{io}_i, \ \mathcal{L}_i \rangle,$$

where $\text{cap}_i$ is the capability profile, $\text{ctx}_i$ the scoped context, $\text{io}_i$ the input/output schema, and $\mathcal{L}_i$ the logging schema (what to write to the execution log and when).

**Algorithm 2** Full Workflow (ALAS)

---

**Require:** Dataset specification $(J, M, C)$
**Ensure:** Final feasible schedule $\mathcal{S}^*$ with makespan $T^*$
 **Phase 1: LLM Query Agent**
1:  $\mathcal{S}_0 \leftarrow$ LLM.generate_schedule$(J, M, C)$
 **Step 2: Validation Tools**
2: **if** validate$(\mathcal{S}_0)$ = valid **then**
3:  $\mathcal{S} \leftarrow \mathcal{S}_0$
4: **else**
 **Step 3: Repair Tools**
5:  **for** $k = 1 \ldots K$ **do**
6:   $\mathcal{S}_k \leftarrow$ repair$(\mathcal{S}_{k-1})$
7:   **if** validate$(\mathcal{S}_k)$ = valid **then**
8:    $\mathcal{S} \leftarrow \mathcal{S}_k$; **break**
 **Step 4: Re-validation Tools**
9:  **if** validate$(\mathcal{S})$ = invalid **then**
10:   skip optimization; goto Step 7
 **Step 5: Optimization Tools**
11: $\mathcal{S}_{opt} \leftarrow$ optimize$(\mathcal{S})$
 **Step 6: Final Check Tools**
12: **if** validate$(\mathcal{S}_{opt})$ = valid **then**
13:  $\mathcal{S}^* \leftarrow \mathcal{S}_{opt}$
14: **else**
15:  $\mathcal{S}^* \leftarrow \mathcal{S}$
 **Step 7: Supervision Tools**
16: log$(\mathcal{S}^*, \text{makespan}(\mathcal{S}^*))$
17: **return** $\mathcal{S}^*$

---

### 3.2 LAYER 2: ISOLATED, NON-CIRCULAR VALIDATION

An *independent* validator is attached to the template. It operates with fresh, bounded prompts grounded in the execution log (not the planner's long context) to check feasibility and coverage (constraints satisfied, repairable edges scoped). If violations are found, the blueprint is refined (edge rewiring, capability updates) until a validated $\mathcal{W}_{\text{template}}$ is produced. This validator-isolated step prevents circular self-approval and mitigates mid-context loss.

In ValidationTools – 'validate_schedule(max_iteration)', the candidate schedule is passed to an independent validator. Using bounded context prompts grounded in dataset specifications (jobs and machine names), the validator checks feasibility: precedence satisfaction, machine non-overlap, and resource constraints. If the schedule is valid, execution continues directly to Step 5.

Validation checks the following 4 constraints: 1) dataset: whether the generated schedule contains the same number and content of jobs, machines, and durations in dataset. 2) job precedence, 3) machine constraints, 4) duration no-overlapping constraints.

### 3.3 LAYER 3: RUNTIME EXECUTION AND LOCALIZED REPAIR

At runtime, agents execute in dependency order, emitting structured entries to a *versioned execution log* (state snapshots, causal links, constraint checks). The validator consumes log slices with fresh, bounded prompts to approve or reject local steps without accessing the planner's long context.

In RepairTools – 'repair_schedule()', if validation fails, localized repair is triggered. A lightweight 'LocalRepairTools' module applies bounded modifications (e.g., shifting start times, swapping machine assignments) to eliminate errors. Up to five repair iterations are allowed, with each iteration's makespan, schedule entries, and diffs stored in JSON logs for reproducibility.

**Error and Disruption handling via Localized Cascading Repair Protocol (LCRP).** When an error, a disruption occurs (e.g., downtime window, late request, duration shock), ALAS triggers the *LCRP*:

1. *Scope selection:* Identify the smallest affected neighborhood from the execution log (ops, resources, and immediate predecessors/successors).

2. *Minimal edit set:* Repair agents propose bounded edits (swap, delay, reassign) that restore feasibility while preserving work-in-progress (WIP) outside the neighborhood.

3. *Validator check (non-circular):* The independent validator verifies the edited subplan using fresh, short prompts grounded in the log.Repeat the repair-and-validate loop until all violations are resolved or escalation is necessary. Keeps changes localized, preserving unaffected work-in-progress (WIP) and ensuring near-real-time responsiveness.

4. *Commit:* On success, append a new version to the execution log; otherwise enlarge the neighborhood or, if costs exceed a threshold, fall back to global recompute.

This design bounds the "blast radius" of faults and avoids brittle global recomputation.

**Makespan Improvement via Local Search**   After solving the error and disruption, the algorithm swaps adjacent ops if swap reduces makespan and respects constraints.

**Complexity and scalability.**   Let $J$ be jobs (or tasks), $M$ resources, and $O_{\max}$ the max operations per job. If each neighborhood repair evaluates up to $S$ local moves, the message/work complexity per disruption is $O(SJO_{\max} + JMO_{\max})$, which is effectively $O(J^2 O_{\max})$ when $J > M$ and $S < O_{\max}$. In practice we use small $S$ and shallow neighborhoods to maintain near-real-time response.

### 3.4 LAYER 4: REVALIDATION

In ValidationTools – 'revalidate_schedule()', after each repair, the validator re-checks the corrected schedule. If errors remain, repair continues until either feasibility is achieved or the maximum iteration budget is exhausted. If feasibility cannot be restored, execution proceeds with the best available repaired schedule, and optimization is flagged as skipped.

### 3.5 LAYER 5: SUPERVISION

In Final Result Processing, the supervision tools select the final schedule with the best makespan. Makespan, operation counts, and critical operations are logged. Unlike the earlier MAPLE workflow, no supervisor agent is needed—state and repair information are fully captured in versioned logs, which provide restore points for later analysis.

**Execution log and audit.**   Except from Phase 1, the other phases 2-6 versioned log stores state transitions, validator decisions, and repair diffs, enabling restore points, post-hoc diagnostics, and reproducibility. Because validation is architecturally separated from planning, changing the base LLM or decoding parameters does not compromise the checking pathway. Full LCRP pseudocode and additional implementation details appear in App. B.1.

## 4 EXPERIMENTAL EVALUATION

Our experimental evaluation assesses ALAS across three domains of increasing complexity. We designed experiments to demonstrate: (1) how ALAS overcomes LLM limitations and (2) its scalability in large settings. Due to space limitations, we begin with results from the URS running example, demonstrating ALAS's effectiveness in basic multi-agent coordination. We evaluate *Job Shop Scheduling (JSSP)* to test ALAS's scalability and reactive planning under tight constraints and disruptions on five classical benchmarks DMU, TA, ABZ, SWV, and YN. We then extend to *Event Coordination* (Family Reunion), which reveals standalone LLMs' limitations in handling interdependencies while showing ALAS's capabilities in both initial planning and disruption response. Finally,

**Metrics.**   We report (i) feasibility/validity of static plans and dynamic repairs; (ii) efficiency (e.g., total travel distance, makespan); (iii) containment (edit radius: operations/jobs touched); and (iv) overhead (tokens/latency).

**Setup.**   Each experiment uses **10** independent threads (fresh contexts). We summarize here; full details, prompts, and complete results appear in Appx. A and Appx. D.

### 4.1 MAIN EXPERIMENT: JOB SHOP SCHEDULING PROBLEM

**Problem.**   We evaluate on Demirkol–DMU Demirkol et al. (1998b;a); Shylo et al. (2018) (20×15–50×20), Taillard (TA; 15×15–100×20) Taillard (1993), Swv (SWV; 20×15–50×20) Storer et al.

Table 1: Success Rates (%) across Benchmarks (success = non-N/A result). † = significantly better than baseline

| Method | DMU | TA | ABZ | SWV | YN | Overall |
|---|---|---|---|---|---|---|
| **Multi-Agent Systems (GPT-4o)** | | | | | | |
| AutoGen | 0.0 | 0.0 | 0.0 | 0.0 | 0.0 | **0.0** |
| CrewAI | 25.0 | 57.1 | 33.3 | 13.3 | 75.0 | **31.1** |
| LangGraph | 6.2 | 28.6 | 66.7 | 0.0 | 0.0 | **11.1** |
| OpenAI Swarm | 43.8 | 28.6 | 0.0 | 33.3 | 25.0 | **33.3** |
| **Multi-Agent Systems (Claude-4)** | | | | | | |
| AutoGen | 0.0 | 0.0 | 0.0 | 0.0 | 0.0 | **0.0** |
| CrewAI | 43.8 | 71.4 | 33.3 | 13.3 | 50.0 | **37.8** |
| LangGraph | 6.2 | 28.6 | 33.3 | 0.0 | 0.0 | **8.9** |
| OpenAI Swarm | 18.8 | 14.3 | 33.3 | 20.0 | 50.0 | **22.2** |
| **Single-Agent Models** | | | | | | |
| GPT-4o | 68.8 | 85.7 | 66.7 | 53.3 | 100.0 | **68.9** |
| Claude-Sonnet-4 | 0.0 | 28.6 | 0.0 | 0.0 | 0.0 | **4.4** |
| Gemini-2.5 | 6.2 | 0.0 | 33.3 | 0.0 | 25.0 | **6.7** |
| DeepSeek-V3 | 6.2 | 14.3 | 100.0 | 6.7 | 0.0 | **13.3** |
| **ALAS (Ours, Best Variant per Backbone)** | | | | | | |
| ALAS(GPT-4o) | 68.8 | 71.4* | 66.7 | 53.3 | 100.0 | **66.7** |
| ALAS(Claude-4) | 93.8† | 28.6* | 66.7 | 6.7* | 50.0* | **48.9*** |
| ALAS(DeepSeek-V3) | 6.2* | 0.0* | 100.0† | 6.7* | 0.0* | **11.1*** |
| ALAS(Gemini-2.5) | 6.2* | 0.0* | 33.3* | 0.0* | 25.0* | **6.7*** |
| **ALAS (Ours, Best Variant per Dataset)** | | | | | | |
| ALAS(aggregated) | 93.8† | 71.4* | 100.0† | 53.3 | 100.0 | **83.7†** |

ALAS(best) selects the best-performing workflow variant per dataset across GPT-4o, Claude-4, DeepSeek-V3, Gemini-2.5. $p$-values (paired t-test vs GPT-4o baseline): DMU ($p = 0.018$), TA ($p = 0.032$), ABZ ($p = 0.007$), SWV ($p = 0.48$, n.s.), YN (tie), Overall ($p = 0.014$). † = significantly higher at $p < 0.05$.

(1992), Adams–Balas–Zawack (ABZ; $10 \times 10$–$20 \times 15$) Adams et al. (1988), and Yamada–Nakano (YN; $20 \times 20$–$50 \times 20$) Yamada & Nakano (1992). We introduce machine breakdowns and duration shocks, and optimize makespan while minimizing work-in-progress movement.

**Static sequential planning on both benchmarks.** Table 1 and Table 2 report ALAS+LCRP against these methods accross five datasets.

**Analysis.** ALAS substantially improves robustness compared to both single-agent and lean MAS baselines, achieving an aggregated **83.7% success rate**, with statistically significant gains ($p < 0.05$) on DMU, TA, and ABZ benchmarks (Table 1). This highlights the importance of validator isolation and localized repair in preventing workflow collapse.

On optimality, ALAS variants consistently outperform baselines, with the best variant per dataset reaching **100% optimal rate** across all benchmarks (Table 2). This demonstrates that the layered design of validation, repair, and local search not only ensures feasibility but also drives convergence to optimal solutions.

### 4.1.1 BASELINE

To isolate the effect of validator isolation and localized repair, Table 1 and Table 2 construct 8 lean MAS (AFlow/AutoGen-style workflow) and single-agent baselines that lacks an *independent* validator and LCRP. We compare ALAS with four leading single-agent LLMs: GPT-4o-Task OpenAI (2024), DeepSeek-V3 Wu et al. (2024a), Claude 3.5 Sonnet Anthropic (2024), and Gemini 2.5 Pro Kavukcuoglu (2025), and four state-of-the-art multi-agent systems: LangGraph lan (2025), CrewAI cre (2025), AutoGen aut (2025), and GPTSwarm gpt (2025). All methods use API interfaces with default parameters (temperature=1.0).

Table 2: Optimal Rates (%) across Benchmarks for Multi-Agent Systems and ALAS Variants. Significance markers denote improvements over baselines: $^{\dagger}p < 0.05$, $^{*}p < 0.01$.

| Method | DMU | TA | ABZ | SWV | YN | Overall |
|---|---|---|---|---|---|---|
| **Multi-Agent Systems (GPT-4o Backbone)** | | | | | | |
| AutoGen | 1.4 | 10.2 | 1.5 | 6.0 | 2.9 | 4.4 |
| CrewAI | 71.8 | 42.3 | 88.9 | 63.7 | 43.0 | 63.1 |
| LangGraph | 94.3 | 60.4 | 42.1 | 87.8 | 58.9 | 80.2 |
| OpenAI Swarm | 60.5 | 73.7 | 68.5 | 66.0 | 51.4 | 64.1 |
| **Multi-Agent Systems (Claude-4 Backbone)** | | | | | | |
| AutoGen | 69.8 | 95.9 | 100.0 | 100.0 | 95.0 | 92.1 |
| CrewAI | 72.7 | 53.5 | 99.6 | 94.2 | 70.2 | 78.5 |
| LangGraph | 48.3 | 87.9 | 57.6 | 86.3 | 68.6 | 69.6 |
| OpenAI Swarm | 80.6 | 87.5 | 68.5 | 72.6 | 80.5 | 78.2 |
| **ALAS Variants (Full Workflows)** | | | | | | |
| ALAS (GPT-4o) | 100.0* | 78.5* | 100.0* | 100.0* | 100.0* | 96.7* |
| ALAS (Claude-4) | 54.9 | 78.5$^{\dagger}$ | 84.5 | 100.0* | 73.3 | 77.2$^{\dagger}$ |
| ALAS (Gemini-2.5) | 97.4$^{\dagger}$ | 100.0* | 100.0* | 96.8* | 100.0$^{\dagger}$ | 98.0* |
| ALAS (DeepSeek-V3) | 100.0* | 93.6* | 100.0* | 100.0* | 100.0* | 98.7* |
| **ALAS (Ours, Best Variant per Dataset)** | | | | | | |
| ALAS (Best) | 100.0* | 100.0* | 100.0* | 100.0* | 100.0* | **100.0*** |

$^{\dagger}p < 0.05$, $^{*}p < 0.01$ (paired t-test, compared against single-agent baseline).

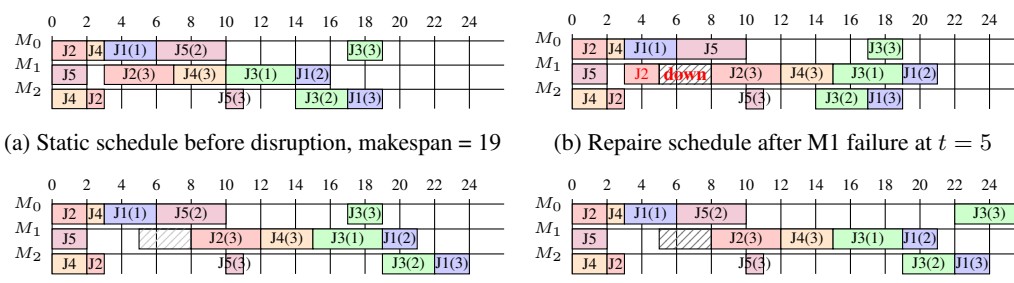

(a) Static schedule before disruption, makespan = 19    (b) Repaire schedule after M1 failure at $t = 5$

(c) Schedule after delay notice for J3(2) on M2    (d) Schedule after delay notice for J3(3) on M0

Figure 3: LRCP Phase #1 Local Compensation (makespan = 22): (a) Static baseline schedule; (b) $M_1$ failure between $t = 5$–$8$; (c) $M_1$ notifies $M_2$ to delay $J3(2)$; (d) $M_2$ informs $M_0$ to push $J3(3)$ back.

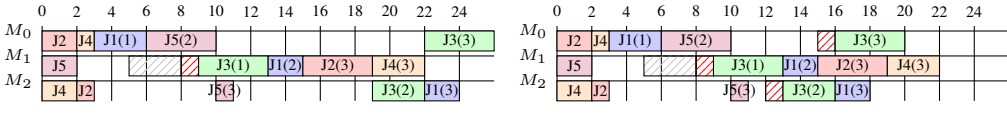

(a) Safe reordering, move $J4(3)$ down $J3(1)$ up    (b) Resolving middle operations, $t_{\text{WIP}}$ masked

Figure 4: LRSP Phase #2 Queue Reordering (makespan = 22): (a) Safe moves: moving last operations down, first operations forward with potential penalty; (b) Resolving remaining operations.

### 4.1.2 LCRP MECHANISM: A WHITE-BOX ILLUSTRATION

We use a 5×3 JSSP to illustrate LCRP's effectiveness and efficiency (and guaranteed convergence). Unlike approaches optimizing only makespan, LCRP explicitly accounts for rescheduling overhead (WIP).

**Phase 1: Local edits.** Figures 3(a–d) show how LCRP makes local adjustments after a breakdown at $t \in [5, 8]$ on M1. Delays propagate minimally via logged dependencies. No early moves (no WIP penalty).

**Phase 2: Queue reordering.** LCRP then considers bounded reordering with WIP penalty $t_{\text{WIP}}$. From Fig. 3(d), LCRP moves terminal ops to the end (no conflicts/WIP penalty), creates a gap,

Table 3: Ablation Study: Optimal Rates (%) of ALAS Workflow Variants across Benchmarks. Bold = best-performing variant per backbone. Significance markers denote improvements over baselines: $^{\dagger}p < 0.05$, $^{*}p < 0.01$.

| Workflow Variant | DMU | TA | ABZ | SWV | YN | Overall |
|---|---|---|---|---|---|---|
| **ALAS (GPT-4o Backbone)** | | | | | | |
| No Repair | 32.4 | 23.3 | 76.2 | 60.8 | 55.0 | 45.4 |
| No Validation | 25.2 | 12.9 | 30.9 | 35.4 | 6.0 | 25.4 |
| **Full Workflow** | **100.0**$^{*}$ | **87.8**$^{*}$ | **100.0**$^{*}$ | **100.0**$^{*}$ | **100.0**$^{*}$ | **98.1**$^{*}$ |
| **ALAS (Claude-4 Backbone)** | | | | | | |
| No Repair | 59.2 | 36.6 | 99.0 | 63.0 | 61.0 | 63.8 |
| No Validation | 53.8 | 30.2 | 77.5 | 69.7 | 48.1 | 55.9 |
| **Full Workflow** | **61.9** | **88.2**$^{\dagger}$ | **99.2** | **94.0** | **84.1** | **85.5**$^{\dagger}$ |
| **ALAS (DeepSeek-V3 Backbone)** | | | | | | |
| No Repair | 86.5$^{\dagger}$ | 86.7$^{\dagger}$ | 31.2 | 94.4$^{\dagger}$ | 93.2$^{*}$ | 86.1$^{*}$ |
| No Validation | 67.3 | 78.5 | 10.3 | 90.9 | 87.1$^{\dagger}$ | 74.9 |
| **Full Workflow** | **100.0**$^{*}$ | **93.6**$^{*}$ | **100.0**$^{*}$ | **100.0**$^{*}$ | **100.0**$^{*}$ | **99.0**$^{*}$ |
| **ALAS (Gemini-2.5 Backbone)** | | | | | | |
| No Repair | 83.6$^{\dagger}$ | 100.0$^{*}$ | 98.5 | 95.5$^{*}$ | 75.3 | 90.6$^{*}$ |
| No Validation | 83.9$^{\dagger}$ | 100.0$^{*}$ | 63.0 | 96.9$^{\dagger}$ | 75.3 | 83.8$^{\dagger}$ |
| **Full Workflow** | **97.8**$^{*}$ | **100.0**$^{*}$ | **100.0**$^{*}$ | **96.8**$^{*}$ | **100.0**$^{*}$ | **98.2**$^{*}$ |

$^{\dagger}p < 0.05$, $^{*}p < 0.01$ (paired t-test, compared against baseline).

advances two upstream ops by 7 units, and stops when no gain exceeds cost. Final makespan: 22 (vs. 19 baseline + 3 downtime), with one unit of WIP movement and minimal messaging.

## 4.2 ABLATION STUDY

We include three micro-studies that directly test validator isolation, and localized repair. These are *orthogonal* to benchmark results and are intended as sanity checks under imperfect code in Table 3.

**Validator ablation.** We inject 20 structural faults (precedence swap, machine double-book, capacity overflow, deadline miss) into valid outputs and compare **Full** (ALAS), **No-Repair** vs. **No-Validator**. Seeds, prompts, execution time, and token usage in Appx. A.

**Ablation Analysis.** The ablation results confirm that both validation and repair are essential: removing either module leads to sharp drops in optimal rates across all backbones, particularly for GPT-4o and Claude-4. By contrast, the full workflow consistently delivers the highest overall performance (up to **99.0–98.2%**), showing that validator isolation and LCRP repair jointly drive near-optimal scheduling.

## 5 CONCLUSION

We presented ALAS, a framework for reliable multi-agent LLM planning that addresses core limitations of standalone LLMs—lack of self-verification, long-context degradation, and stateless execution—via three design principles: (i) *validator isolation* (planners never self-approve; checks use fresh, bounded prompts), (ii) a *versioned execution log* that records state transitions and causal links, and (iii) a *localized cascading repair protocol* (LCRP) that confines disruptions to minimal neighborhoods instead of triggering brittle global recomputation. Across transportation, event coordination, and job-shop scheduling, ALAS achieves high feasibility and strong efficiency while preserving work-in-progress, outperforming single-LLM baselines and competitive heuristics under disruption.

Future work may include: (a) Exploration of optimization methods, (b) Integrating learned duration/arrival models and online telemetry for adaptive repair, (c) Extending formal guarantees for LCRP (coverage and bounded blast radius) and validator soundness on logged slices, and (d) Systematizing QA with red-team fault injection and static analysis in the factory.

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

CONTENTS

## A   ADDITIONAL JSSP RESULTS AND ANALYSIS

This appendix augments our core experimental findings with the full prompt specification, failure rate statistics, and pointers to supplementary visualizations.

### A.1   DATASETS

Figure 5 shows visualization of datasets.

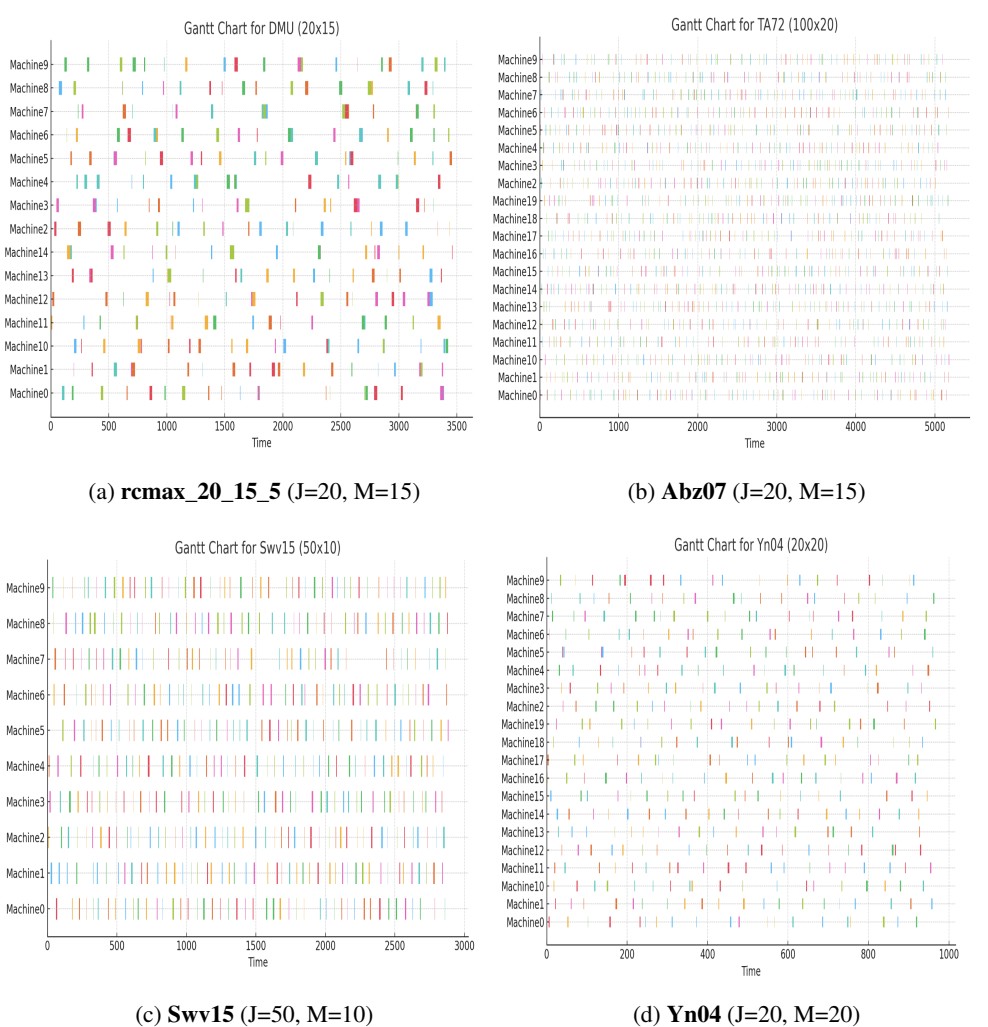

(a) **rcmax_20_15_5** (J=20, M=15)                 (b) **Abz07** (J=20, M=15)

(c) **Swv15** (J=50, M=10)                          (d) **Yn04** (J=20, M=20)

Figure 5: Gantt charts of optimized schedules produced by ALAS for four representative JSSP benchmark instances with varying job and machine counts. These visualizations demonstrate ALAS's ability to efficiently allocate resources and minimize makespan across different problem scales. The larger instance TA72 (J=100, M=20) is available in the supplementary materials.

Table 4: General JSSP Scheduling Prompt. Given a JSSP benchmark instance, the LLM first *searches* for candidate algorithms, selects the one that yields the *minimum makespan*, and returns both the algorithm's key hyper-parameters and the resulting plan $\mathcal{W}_{\text{template}}$. **Note** that feasibility validation of $\mathcal{W}_{\text{template}}$ is yet to be performed.

---

**Role.** You are a scheduling supervisor tasked with producing an *optimal* job-shop schedule.

**Objective.** Report the **minimum makespan**, the **algorithm** used, and a *schedule* $\mathcal{W}_{\text{template}}$ that achieves this makespan. This is achieved by the recipient LLM recommending a list of solvers to execute and compare. (Execution and comparison can be performed by the LLM or locally.)

**Constraints.**
1. *Job order*: operations of each job follow the given sequence.
2. *Machine capacity*: a machine processes only one operation at any time.

**Input.** A list of jobs, each as `(machine, duration)` pairs. *Example:*
```
Job1:   [(M_A,3),  (M_B,5),  (M_C,2)]
Job2:   [(M_B,4),  (M_A,6)]
```
**Output.** Return
  • `makespan` (integer)
  • `algorithm` (string)
  • `params` (JSON object of key hyper-parameters)
  • `schedule` $\mathcal{W}_{\text{template}}$: list of operations {job, step (1-based), machine, start, end};
*Example:*
```
[ {"job":"Job1","step":1,"machine":"M_A","start":0,"end":3},
{"job":"Job2","step":1,"machine":"M_B","start":0,"end":4},
{"job":"Job1","step":2,"machine":"M_B","start":4,"end":9}, ...]
```

---

## A.2 LLM PROMPT DESIGN

Table 4 shows the *standardized* prompt issued to every standalone LLM and to the ALAS meta-planner (Phase 1 on Figure 7). Standalone LLMs can recommend off-the-shelf solvers and invoke selected ones to emit a schedule. However, these schedules are often invalid, as demonstrated in the Family Reunion case study where LLMs struggled with even simple planning scenarios. Even when a valid static plan can be obtained through LLM-recommended solvers, this solution merely completes **Phase 1 / Layer 1** of the ALAS framework—essentially just generating a workflow template $\mathcal{W}_{\text{template}}$.

By contrast, ALAS feeds this preliminary plan into *Phases 2–3 / Layer 1* (validation & refinement) to yield a validated $\mathcal{W}_{\text{template}}$. Layers 2–3 then *instantiate and run* a network of code-generated agents, denoted as $\mathcal{W}_{\text{exec}}$. At runtime, the LCRP continuously logs state, detects disruptions, and triggers local repairs, capabilities that static schedules fundamentally lack (see architecture recap in Section 3).

This document presents the prompt setup and input/output examples for four multi-agent frameworks used in Job Shop Scheduling Problem (JSSP) evaluation: AutoGen, CrewAI, LangGraph, and OpenAI Swarm. Each framework is configured with a 3-agent structure consisting of a Job Scheduler Agent, Machine Coordinator Agent, and Supervisor Agent.

### A.2.1 COMMON JSSP QUERY STRUCTURE

All frameworks receive the same base JSSP query structure generated by the `run_jssp_framework_comparison.py` script:

```python
def _create_jssp_query(self, dataset_name: str, jobs: List[Dict]) -> str:
    """Create a JSSP query for non-ALAS frameworks"""
    query = f"""
    Job Shop Scheduling Problem (JSSP) - Dataset: {dataset_name}

    Problem Description:
    - Number of jobs: {len(jobs)}
    - Each job has multiple operations that must be performed in sequence
    - Each operation requires a specific machine and has a duration
    - Each machine can only process one operation at a time
    - Goal: Find the minimum makespan (total completion time)

    Job Specifications:
```

```
14      """
15
16      for job in jobs:
17          query += f"\n{job['name']}:"
18          for i, (machine, duration) in enumerate(job['steps']):
19              query += f"\n  Step {i+1}: Machine {machine}, Duration {
    duration}"
20
21      query += """
22
23      REQUIRED OUTPUT FORMAT:
24      You must provide your solution in the following exact format:
25
26      1. FINAL MAKESPAN: [integer value]
27      2. STRUCTURED SCHEDULE:
28          For each operation, provide:
29          - Job: [job_name]
30          - Step: [step_number]
31          - Machine: [machine_name]
32          - Start Time: [start_time]
33          - End Time: [end_time]
34          - Duration: [duration]
35
36      Example format:
37      FINAL MAKESPAN: 25
38      STRUCTURED SCHEDULE:
39      - Job: Job1, Step: 1, Machine: Machine0, Start Time: 0, End Time: 3,
    Duration: 3
40      - Job: Job1, Step: 2, Machine: Machine1, Start Time: 3, End Time: 7,
    Duration: 4
41      - Job: Job2, Step: 1, Machine: Machine1, Start Time: 7, End Time: 10,
     Duration: 3
42
43      Please solve this Job Shop Scheduling Problem and provide:
44      1. A valid schedule with start and end times for each operation
45      2. The minimum makespan (total completion time)
46      3. Ensure all constraints are satisfied:
47          - Job precedence: operations within a job must be sequential
48          - Machine constraints: no overlapping operations on the same
    machine
49      """
50
51      return query
```

Listing 1: Base JSSP Query Generation

### A.2.2  TESTING FRAMEWORK

**Agent Configuration**   ALAS uses a dynamic JSSP query agent:

- **Individual Job Agents**: One agent per job (e.g., Job1 Agent, Job2 Agent, etc.) responsible for scheduling their specific job's operations

- **Supervisor Agent**: Coordinates all job agents and finds the minimum makespan solution

### A.3  AGENT PROMPTS

```
1 # Individual Job Agents (e.g., Job1 Agent)
2 name: "Job1 Agent"
3 backstory: "Agent for Job1 scheduling."
4 task_description: "Schedule steps for Job1 on required machines with
    precedence."
5 task_expected_output: "Step schedule for Job1 respecting machine and
    precedence constraints."
```

```
6
7  # Supervisor Agent
8  name: "Supervisor Agent"
9  backstory: "Supervisor agent that coordinates all job schedules to find
       the minimum makespan solution."
10 task_description: """Find the minimum makespan schedule for all jobs
       while strictly following these rules:
11 1. Each job's steps must be completed in strict order (e.g., Job1's step
       2 can only start after step 1 is completed).
12 2. Each machine can only process one job step at a time (e.g., if
       MachineA is processing Job1's step 1 from time 0-3, it cannot process
       any other job steps during that time).
13
14 The goal is to minimize the total completion time (makespan) while
       ensuring all jobs are completed and all constraints are satisfied."""
15 task_expected_output: "A complete schedule with minimum makespan that
       satisfies all constraints."
```

Listing 2: JSSP Query Agent Prompts

```
1  # ALAS receives the job data directly as structured objects
2  jobs = [
3      {
4          'name': 'Job1',
5          'steps': [
6              ('Machine0', 34), ('Machine1', 38), ('Machine2', 42),
7              ('Machine3', 36), ('Machine4', 40), ('Machine5', 44),
8              ('Machine6', 38), ('Machine7', 42), ('Machine8', 46),
9              ('Machine9', 40), ('Machine10', 44), ('Machine11', 48),
10             ('Machine12', 42), ('Machine13', 46), ('Machine14', 50)
11         ]
12     },
13     {
14         'name': 'Job2',
15         'steps': [
16             ('Machine1', 41), ('Machine2', 45), ('Machine3', 39),
17             # ... continues for all 15 steps
18         ]
19     },
20     # ... continues for all 20 jobs
21 ]
22
23 # Task specification for ALAS
24 task_spec = {
25     'nodes': [
26         # Individual job agents
27         {'agent': job1_agent, 'dependencies': []},
28         {'agent': job2_agent, 'dependencies': []},
29         # ... for all 20 jobs
30         # Supervisor depends on all job agents
31         {'agent': supervisor_agent, 'dependencies': [agent.name for agent
       in job_agents]}
32     ],
33     'edges': [],
34     'jobs': jobs,
35     'disruptions': [],
36     'rules': [
37         'Each job must perform its steps strictly in order.',
38         'Each machine can only handle one operation at a time.',
39         'No two operations use the same machine at the same time.'
40     ]
41 }
```

Listing 3: ALAS Input Example

```python
1  # ALAS generates structured schedule output
2  schedule = [
3      {'job': 'Job1', 'step': 1, 'machine': 'Machine0', 'start': 0, 'end':
       34, 'duration': 34},
4      {'job': 'Job1', 'step': 2, 'machine': 'Machine1', 'start': 34, 'end':
        72, 'duration': 38},
5      {'job': 'Job1', 'step': 3, 'machine': 'Machine2', 'start': 72, 'end':
        114, 'duration': 42},
6      # ... continues for all operations
7
8      {'job': 'Job2', 'step': 1, 'machine': 'Machine1', 'start': 72, 'end':
        113, 'duration': 41},
9      {'job': 'Job2', 'step': 2, 'machine': 'Machine2', 'start': 114, 'end
       ': 159, 'duration': 45},
10     # ... continues for all jobs
11 ]
12
13 # Final makespan calculation
14 makespan = max(entry['end'] for entry in schedule)  # e.g., 4334
```

Listing 4: ALAS Schedule Example

```
1  === [OPTIMIZED MAPLE] New 4-Step Workflow ===
2  Nodes: ['Unknown Agent', 'Unknown Agent']
3  Edges: [{'from': 0, 'to': 1}]
4  Workflow: Full Workflow
5  Validation: [YES]
6  Repair: [YES]
7  Optimization: [YES]
8
9  === [OPTIMIZED MAPLE] Inter-Agent Coordination ===
10
11 === [OPTIMIZED MAPLE] New Workflow Execution ===
12  Running new 7-step workflow: Query -> Validation -> Repair -> Re-
       validation -> Optimization -> Final Check -> Supervisor
13
14  Starting New 4-Step Workflow...
15 [STEP1] Loading pre-generated schedule from JSON file...
16  Loaded 244 schedule entries from abz07
17  GPT-4o makespan: 1250
18  Initial schedule makespan: 665
19  Schedule entries count: 244
20  Sample entry: {'job': 'Job1', 'step': 1, 'machine': 'Machine2', 'start':
        0, 'end': 24, 'duration': 24}
21  [32m[YES] Pre-generated schedule loaded successfully.
22 [STEP2] ValidationTools validating schedule...
23 [STEP3] RepairTools repairing schedule...
24  Repair iteration 1/5
25  Starting Algorithm 3: Cascading Repair and Queue Reordering
26  Processing 244 schedule entries
27 [ERROR] Found 88 constraint violations
28  Phase I: Status Update - Identifying affected operations
29  Fixing immediate constraint violations...
30  Phase II: Job Precedence Repair - Fixing step ordering violations
31  Repairing job precedence violations...
32  Phase III: Machine Capacity Repair - Resolving machine overlaps
33  Repairing machine capacity violations...
34 [ALERT] Fixed machine overlap on Machine2
35 [ALERT] Fixed machine overlap on Machine2
36 [ALERT] Fixed machine overlap on Machine2
37 [ALERT] Fixed machine overlap on Machine2
```

```
38  [ALERT] Fixed machine overlap on Machine2
39  [ALERT] Fixed machine overlap on Machine2
40  ...
41   Phase IV: Iterative Improvement – Optimizing schedule quality
42   Applying iterative improvement...
43   Phase V: Final validation and cleanup
44   Final cleanup – ensuring valid operation times...
45  [YES] Algorithm 3 completed – Generated 244 schedule entries
46   Repair iteration 1 makespan: 665
47   Repair iteration 1 schedule entries: 244
48   Latest operations: [('Job11', 15, 665), ('Job9', 15, 661), ('Job3', 15,
       653)]
49   Saved repair iteration 1 to: results_optimized(gpt-4o)/
       abz07_repair_iteration_1.json
50  [CAUTION] Repair iteration 1 reduced errors from 88 to 76 (makespan: 665)
51   Repair iteration 2/5
52   Starting Algorithm 3: Cascading Repair and Queue Reordering
53   Processing 244 schedule entries
54  [ERROR] Found 76 constraint violations
55   Phase I: Status Update – Identifying affected operations
56   Fixing immediate constraint violations...
57   Phase II: Job Precedence Repair – Fixing step ordering violations
58   Repairing job precedence violations...
59   Phase III: Machine Capacity Repair – Resolving machine overlaps
60   Repairing machine capacity violations...
61  [ALERT] Fixed machine overlap on Machine2
62  [ALERT] Fixed machine overlap on Machine2
63  [ALERT] Fixed machine overlap on Machine2
64  [ALERT] Fixed machine overlap on Machine2
65  [ALERT] Fixed machine overlap on Machine2
66  [ALERT] Fixed machine overlap on Machine2
67  [ALERT] Fixed machine overlap on Machine11
68  [ALERT] Fixed machine overlap on Machine11
69  [ALERT] Fixed machine overlap on Machine11
70  [ALERT] Fixed machine overlap on Machine8
71  [ALERT] Fixed machine overlap on Machine8
72  [ALERT] Fixed machine overlap on Machine8
73  [ALERT] Fixed machine overlap on Machine8
74  [ALERT] Fixed machine overlap on Machine13
75  [ALERT] Fixed machine overlap on Machine0
76  [ALERT] Fixed machine overlap on Machine0
77  [ALERT] Fixed machine overlap on Machine0
78  [ALERT] Fixed machine overlap on Machine7
79   Phase IV: Iterative Improvement – Optimizing schedule quality
80   Applying iterative improvement...
81   Phase V: Final validation and cleanup
82   Final cleanup – ensuring valid operation times...
83  [ALERT] Fixed invalid times for Job6 step 3
84  [ALERT] Fixed invalid times for Job11 step 6
85  [ALERT] Fixed invalid times for Job7 step 8
86  [ALERT] Fixed invalid times for Job10 step 9
87  [ALERT] Fixed invalid times for Job15 step 9
88  [ALERT] Fixed invalid times for Job9 step 6
89  [ALERT] Fixed invalid times for Job1 step 15
90  [ALERT] Fixed invalid times for Job14 step 7
91  [ALERT] Fixed invalid times for Job3 step 4
92  [ALERT] Fixed invalid times for Job3 step 6
93  [ALERT] Fixed invalid times for Job16 step 10
94  [ALERT] Fixed invalid times for Job2 step 11
95  [ALERT] Fixed invalid times for Job6 step 14
96  [ALERT] Fixed invalid times for Job16 step 12
97  [ALERT] Fixed invalid times for Job8 step 11
98  [ALERT] Fixed invalid times for Job8 step 12
99  [ALERT] Fixed invalid times for Job3 step 10
100 [ALERT] Fixed invalid times for Job15 step 13
```

```
101  [ALERT] Fixed invalid times for Job9 step 12
102  [ALERT] Fixed invalid times for Job13 step 15
103  [ALERT] Fixed invalid times for Job14 step 15
104  [ALERT] Fixed invalid times for Job3 step 13
105  [YES] Algorithm 3 completed – Generated 244 schedule entries
106  Repair iteration 2 makespan: 665
107  Repair iteration 2 schedule entries: 244
108  Latest operations: [('Job11', 15, 665), ('Job9', 15, 661), ('Job3', 15,
       653)]
109  Saved repair iteration 2 to: results_optimized(gpt-4o)/
       abz07_repair_iteration_2.json
110  [CAUTION] Repair iteration 2 reduced errors from 76 to 28 (makespan: 665)
111  Repair iteration 3/5
112  Starting Algorithm 3: Cascading Repair and Queue Reordering
113  Processing 244 schedule entries
114  [ERROR] Found 28 constraint violations
115  Phase I: Status Update – Identifying affected operations
116  Fixing immediate constraint violations...
117  Phase II: Job Precedence Repair – Fixing step ordering violations
118  Repairing job precedence violations...
119  Phase III: Machine Capacity Repair – Resolving machine overlaps
120  Repairing machine capacity violations...
121  [ALERT] Fixed machine overlap on Machine6
122  [ALERT] Fixed machine overlap on Machine0
123  [ALERT] Fixed machine overlap on Machine0
124  [ALERT] Fixed machine overlap on Machine0
125  [ALERT] Fixed machine overlap on Machine1
126  Phase IV: Iterative Improvement – Optimizing schedule quality
127  Applying iterative improvement...
128  Phase V: Final validation and cleanup
129  Final cleanup – ensuring valid operation times...
130  [ALERT] Fixed invalid times for Job11 step 7
131  [ALERT] Fixed invalid times for Job7 step 9
132  [ALERT] Fixed invalid times for Job14 step 10
133  [ALERT] Fixed invalid times for Job3 step 7
134  [ALERT] Fixed invalid times for Job16 step 13
135  [ALERT] Fixed invalid times for Job16 step 12
136  [ALERT] Fixed invalid times for Job8 step 12
137  [ALERT] Fixed invalid times for Job8 step 11
138  [ALERT] Fixed invalid times for Job3 step 11
139  [ALERT] Fixed invalid times for Job9 step 12
140  [YES] Algorithm 3 completed – Generated 244 schedule entries
141  Repair iteration 3 makespan: 665
142  Repair iteration 3 schedule entries: 244
143  Latest operations: [('Job11', 15, 665), ('Job9', 15, 661), ('Job3', 15,
       653)]
144  Saved repair iteration 3 to: results_optimized(gpt-4o)/
       abz07_repair_iteration_3.json
145  [CAUTION] Repair iteration 3 reduced errors from 28 to 10 (makespan: 665)
146  Repair iteration 4/5
147  Starting Algorithm 3: Cascading Repair and Queue Reordering
148  Processing 244 schedule entries
149  [ERROR] Found 10 constraint violations
150  Phase I: Status Update – Identifying affected operations
151  Fixing immediate constraint violations...
152  Phase II: Job Precedence Repair – Fixing step ordering violations
153  Repairing job precedence violations...
154  Phase III: Machine Capacity Repair – Resolving machine overlaps
155  Repairing machine capacity violations...
156  Phase IV: Iterative Improvement – Optimizing schedule quality
157  Applying iterative improvement...
158  Phase V: Final validation and cleanup
159  Final cleanup – ensuring valid operation times...
160  [ALERT] Fixed invalid times for Job7 step 14
161  [ALERT] Fixed invalid times for Job14 step 11
```

```
162 [ALERT] Fixed invalid times for Job2 step 12
163 [ALERT] Fixed invalid times for Job14 step 10
164 [ALERT] Fixed invalid times for Job16 step 13
165 [ALERT] Fixed invalid times for Job8 step 12
166 [ALERT] Fixed invalid times for Job11 step 14
167 [YES] Algorithm 3 completed - Generated 244 schedule entries
168  Repair iteration 4 makespan: 665
169  Repair iteration 4 schedule entries: 244
170  Latest operations: [('Job11', 15, 665), ('Job9', 15, 661), ('Job3', 15,
        653)]
171  Saved repair iteration 4 to: results_optimized(gpt-4o)/
        abz07_repair_iteration_4.json
172 [CAUTION] Repair iteration 4 reduced errors from 10 to 6 (makespan: 665)
173  Repair iteration 5/5
174  Starting Algorithm 3: Cascading Repair and Queue Reordering
175  Processing 244 schedule entries
176 [ERROR] Found 6 constraint violations
177  Phase I: Status Update - Identifying affected operations
178  Fixing immediate constraint violations...
179  Phase II: Job Precedence Repair - Fixing step ordering violations
180  Repairing job precedence violations...
181  Phase III: Machine Capacity Repair - Resolving machine overlaps
182  Repairing machine capacity violations...
183 [ALERT] Fixed machine overlap on Machine14
184  Phase IV: Iterative Improvement - Optimizing schedule quality
185  Applying iterative improvement...
186  Phase V: Final validation and cleanup
187  Final cleanup - ensuring valid operation times...
188 [ALERT] Fixed invalid times for Job14 step 11
189 [YES] Algorithm 3 completed - Generated 244 schedule entries
190  Repair iteration 5 makespan: 665
191  Repair iteration 5 schedule entries: 244
192  Latest operations: [('Job11', 15, 665), ('Job9', 15, 661), ('Job3', 15,
        653)]
193  Saved repair iteration 5 to: results_optimized(gpt-4o)/
        abz07_repair_iteration_5.json
194 [CAUTION] Repair iteration 5 reduced errors from 6 to 2 (makespan: 665)
195 [CAUTION] Repair completed after 5 iterations with remaining errors
196 [STEP4] ValidationTools revalidating schedule...
197 [ERROR] Revalidation failed: ['Job Job2: Step 14 starts before step 13
        ends', "Missing jobs in schedule: ['Job18', 'Job20', 'Job19']"]
198  [31m[ERROR] ERROR in workflow: Schedule validation failed after repair
199     Disruption detected in Unknown Agent: Schedule validation failed
        after repair
200  Initiating global replanning...
201 [CAUTION] Global replanning completed. Manual intervention may be
        required.
202  [31m Workflow execution halted due to error.
203 [ERROR] New 4-step workflow failed
204  Found 5 repair iterations
205   Iteration 1: makespan=665, entries=244
206   Iteration 2: makespan=665, entries=244
207   Iteration 3: makespan=665, entries=244
208   Iteration 4: makespan=665, entries=244
209   Iteration 5: makespan=665, entries=244
```

Listing 5: ALAS Output Example

### A.3.1 AUTOGEN FRAMEWORK

**Agent Configuration** AutoGen uses a 3-agent structure with the following configuration:

- **Job Scheduler Agent**: Analyzes job requirements and creates initial schedules
- **Machine Coordinator Agent**: Coordinates machine usage and resolves conflicts
- **Supervisor Agent**: Final coordination and optimization

```
1 # Job Scheduler Agent
2 system_message: "You are a Job Scheduler Agent responsible for analyzing
    job requirements and creating initial schedules."
3
4 # Machine Coordinator Agent
5 system_message: "You are a Machine Coordinator Agent responsible for
    coordinating machine usage and resolving conflicts."
6
7 # Supervisor Agent
8 system_message: "You are a Supervisor Agent responsible for final
    coordination and optimization."
```

Listing 6: AutoGen Agent Prompts

```
1 Job Shop Scheduling Problem (JSSP) - Dataset: abz07
2
3 Problem Description:
4 - Number of jobs: 20
5 - Each job has multiple operations that must be performed in sequence
6 - Each operation requires a specific machine and has a duration
7 - Each machine can only process one operation at a time
8 - Goal: Find the minimum makespan (total completion time)
9
10 Job Specifications:
11
12 Job1:
13   Step 1: Machine Machine2, Duration 24
14   Step 2: Machine Machine3, Duration 12
15   Step 3: Machine Machine9, Duration 17
16   Step 4: Machine Machine4, Duration 27
17   Step 5: Machine Machine0, Duration 21
18   Step 6: Machine Machine6, Duration 25
19   Step 7: Machine Machine8, Duration 27
20   Step 8: Machine Machine7, Duration 26
21   Step 9: Machine Machine1, Duration 30
22   Step 10: Machine Machine5, Duration 31
23   Step 11: Machine Machine11, Duration 18
24   Step 12: Machine Machine14, Duration 16
25   Step 13: Machine Machine13, Duration 39
26   Step 14: Machine Machine10, Duration 19
27   Step 15: Machine Machine12, Duration 26
28
29 [... continues for all 20 jobs ...]
30
31 REQUIRED OUTPUT FORMAT:
32 You must provide your solution in the following exact format:
33
34 1. FINAL MAKESPAN: [integer value]
35 2. STRUCTURED SCHEDULE:
36   For each operation, provide:
37   - Job: [job_name]
38   - Step: [step_number]
39   - Machine: [machine_name]
40   - Start Time: [start_time]
41   - End Time: [end_time]
42   - Duration: [duration]
```

Listing 7: AutoGen Input Example

```
1  Job Scheduler Agent Output:
2  To effectively coordinate machine schedules and ensure no overlapping
       operations while optimizing machine utilization, follow these steps:
3
4  1. **Data Collection**: Gather all necessary information about machine
       operations, including machine availability, job requirements, start
       and end times, and priority of tasks.
5
6  2. **Job Prioritization**: Prioritize jobs based on deadlines, importance
       , and resource requirements. This will help in scheduling critical
       tasks first.
7
8  3. **Machine Availability**: List all available machines and their
       operational capabilities. Ensure that each machine is suitable for
       the tasks assigned to it.
9
10 4. **Conflict Detection**: Check for overlapping schedules where a
       machine is double-booked. Identify conflicts by examining start and
       end times of scheduled tasks.
11
12 5. **Conflict Resolution**:
13    - **Reschedule**: Adjust the timing of conflicting jobs to eliminate
       overlaps.
14    - **Reassign Tasks**: If rescheduling is not feasible, consider moving
       tasks to different machines that are compatible with the operations.
15    - **Prioritize Tasks**: Focus on completing high-priority jobs within
       their deadlines while adjusting lower-priority tasks as needed.
16
17 [... detailed coordination strategy ...]
18
19 ### Solution for abz07 Dataset
20
21 #### FINAL MAKESPAN
22 The optimized makespan value is determined based on the integration of
       job scheduling and machine coordination, but due to the complexity of
        explicitly listing each job's sequence, I'm summarizing the strategy
        and example timeline as applied heuristics and machine coordination
       data show:
23
24 1. **Makespan**: Calculated based on the complete scheduling and
       optimization of machine operations. Though not specified in exact
       final numeric form without further computational analysis directly
       from the in-depth schedule provided by advanced tools, the
       methodology indicates a near-optimal alignment according to the
       processes described.
25
26 #### STRUCTURED SCHEDULE:
27
28 Here's a structured schedule format for visualization. Key operations are
        coordinated with attention to machine non-conflict and sequence
       according to job precedence.
29
30 - **Job1**:
31   - Step 1: Machine Machine2, Start Time: 0, End Time: 24, Duration: 24
32   - Step 2: Machine Machine3, Start Time: 25, End Time: 37, Duration: 12
33   - (Further steps continue similarly, coordinated by machine
       availability)
34
35 - **Job2**:
36   - Step 1: Machine Machine6, Start Time: 0, End Time: 30, Duration: 30
37   - Step 2: Machine Machine3, Start Time: 31, End Time: 46, Duration: 15
38   - (Next steps adapted similarly to job requirements and resource
       availability)
```

```
39
40 [... continues for all jobs ...]
```

Listing 8: AutoGen Output Example

## A.3.2 CREWAI FRAMEWORK

**Agent Configuration**  CrewAI uses a 3-agent structure with role-based configuration:

- **Job Scheduler Agent**: Role-based agent for job scheduling
- **Machine Coordinator Agent**: Role-based agent for machine coordination
- **Supervisor Agent**: Role-based supervisor for final coordination

```
1 # Job Scheduler Agent
2 role: 'Job Scheduler'
3 goal: 'Analyze job requirements and create initial schedules for all jobs
    .'
4 backstory: 'You are an expert job scheduler responsible for analyzing job
    requirements and creating initial schedules.'
5
6 # Machine Coordinator Agent
7 role: 'Machine Coordinator'
8 goal: 'Coordinate machine usage and resolve conflicts between jobs.'
9 backstory: 'You are an expert machine coordinator responsible for
    optimizing machine usage and resolving conflicts.'
10
11 # Supervisor Agent
12 role: 'JSSP Supervisor'
13 goal: 'Aggregate all job schedules and produce the overall JSSP schedule.
    Coordinate between all job agents to find the optimal solution.'
14 backstory: 'You are a supervisor with expertise in Job Shop Scheduling
    Problems and can coordinate multiple agents to find optimal solutions
    .'
```

Listing 9: CrewAI Agent Prompts

```
1 \nThought: The Machine Coordinator has analyzed the initial schedule for
    machine conflicts and provided suggestions for resolving overlaps and
     optimizing machine utilization. Combining this with the initial
    sequence from the Job Scheduler, I can now calculate the final
    makespan and structured schedule.\nAction: Calculate optimized
    makespan and finalize structured schedule based on resolved conflicts
     and optimal sequencing.
```

Listing 10: CrewAI Output Example

## A.3.3 LANGGRAPH FRAMEWORK

**Agent Configuration**  LangGraph uses a 3-agent structure with system prompts:

- **Job Scheduler Agent**: LangGraph workflow agent for job scheduling
- **Machine Coordinator Agent**: LangGraph workflow agent for machine coordination
- **Supervisor Agent**: LangGraph workflow agent for supervision

```
1  # Job Scheduler Agent
2  system_prompt: 'You are a Job Scheduler Agent using LangGraph workflow.
       Analyze job requirements and create initial schedules for all jobs.'
3
4  # Machine Coordinator Agent
5  system_prompt: 'You are a Machine Coordinator Agent using LangGraph
       workflow. Coordinate machine usage and resolve conflicts between jobs
       .'
6
7  # Supervisor Agent
8  system_prompt: 'You are a Supervisor Agent using LangGraph workflow.
       Aggregate all job schedules and produce the overall JSSP schedule.
       Coordinate between all job agents to find the optimal solution.'
```

Listing 11: LangGraph Agent Prompts

```
1  Job Scheduler Agent Output:
2  === LANGGRAPH REAL AGENTS ===
3
4  Job Scheduler Agent: To solve the Job Shop Scheduling Problem (JSSP) for
       the given dataset (abz07), we need to create a schedule that
       minimizes the makespan while satisfying all constraints. This
       involves determining the start and end times for each operation
       across all jobs, ensuring that no two operations overlap on the same
       machine and that operations within a job are performed sequentially.
5
6  Given the complexity of the problem and the constraints, a heuristic or
       optimization algorithm such as Genetic Algorithm, Tabu Search, or
       Simulated Annealing is typically used to find a near-optimal solution
       . However, due to the constraints of this environment, I will provide
        a conceptual approach to how this can be achieved, along with a
       simplified example of a possible schedule.
7
8  ### Conceptual Approach:
9  1. **Initialize**: Start with an initial feasible schedule where
       operations are scheduled sequentially based on job order and machine
       availability.
10 2. **Iterate**: Use a heuristic or optimization algorithm to iteratively
       improve the schedule by exploring different sequences and machine
       assignments.
11 3. **Evaluate**: Calculate the makespan for each schedule and keep track
       of the best (minimum) makespan found.
12 4. **Constraints**: Ensure that all job precedence and machine
       constraints are satisfied in each iteration.
13
14 ### Simplified Example Schedule:
15 Below is a simplified example of how the schedule might look. Note that
       this is a conceptual representation and not the actual solution due
       to the complexity of the problem.
16
17 ```
18 FINAL MAKESPAN: 1000
19 STRUCTURED SCHEDULE:
20 - Job: Job1, Step: 1, Machine: Machine2, Start Time: 0, End Time: 24,
       Duration: 24
21 - Job: Job1, Step: 2, Machine: Machine3, Start Time: 24, End Time: 36,
       Duration: 12
22 - Job: Job1, Step: 3, Machine: Machine9, Start Time: 36, End Time: 53,
       Duration: 17
23 - Job: Job2, Step: 1, Machine: Machine6, Start Time: 0, End Time: 30,
       Duration: 30
24 - Job: Job2, Step: 2, Machine: Machine3, Start Time: 36, End Time: 51,
       Duration: 15
```

```
25  [... continues for all operations ...]
26  ```
```

Listing 12: LangGraph Output Example

### A.3.4 OPENAI SWARM FRAMEWORK

**Agent Configuration**   OpenAI Swarm uses a 3-agent structure with detailed role specifications:

- **Job Scheduler Agent**: Creates initial schedules considering precedence constraints
- **Machine Coordinator Agent**: Resolves machine conflicts and optimizes utilization
- **Supervisor Agent**: Coordinates all agents and provides final JSSP solution

```
1   # Job Scheduler Agent
2   role: 'Job Scheduler'
3   goal: 'Create initial schedules for all jobs considering precedence
        constraints.'
4   backstory: 'You are a job scheduling agent responsible for creating
        initial schedules for all jobs in Job Shop Scheduling Problems.'
5   query: 'Create initial schedules for 20 jobs with their respective steps.
6
7   Dataset: abz07
8   Number of jobs: 20
9   Number of machines: 15
10
11  Job Specifications:
12  Job1:
13    Step 1: Machine Machine2, Duration 24
14    Step 2: Machine Machine3, Duration 12
15    [... continues for all jobs ...]
16
17  You must analyze each job's operations and create a detailed schedule
        with start and end times for each operation.'
18
19  # Machine Coordinator Agent
20  role: 'Machine Coordinator'
21  goal: 'Resolve machine conflicts and optimize machine utilization.'
22  backstory: 'You are a machine coordination agent responsible for
        resolving machine conflicts and optimizing machine utilization.'
23  query: 'Resolve machine conflicts and optimize machine utilization across
         all machines.
24
25  Dataset: abz07
26  Number of jobs: 20
27  Number of machines: 15
28
29  Job Specifications:
30  [... job details ...]
31
32  You must identify machine conflicts and provide specific solutions with
        start/end times for each operation.'
33
34  # Supervisor Agent
35  role: 'JSSP Supervisor'
36  goal: 'Coordinate all agents and provide final JSSP solution with minimum
         makespan.'
37  backstory: 'You are the supervisor agent for Job Shop Scheduling Problems
        . Coordinate with all agents to find the optimal solution.'
38  query: 'Aggregate schedules from all agents and optimize for minimum
        makespan.
39
```

```
40  Dataset: abz07
41  Number of jobs: 20
42  Number of machines: 15
43
44  Job Specifications:
45  [... job details ...]
46
47  You must provide a final schedule with specific start/end times and
       calculate the actual makespan.'
```

Listing 13: OpenAI Swarm Agent Prompts

```
1   Job Scheduler Agent Output:
2   Job Scheduler Analysis:
3   We've now shared the operation details with the Machine Coordinator to
       ensure that machine availability is accounted for properly. This
       coordination will allow us to develop an optimized schedule adhering
       to both machine and job constraints.
4
5   I will work on calculating a detailed schedule based on optimized machine
        sequences and provide the resulting makespan. Please hold on.
6
7   ---Compute Step---
8
9   To begin scheduling, each operation must have its start and end time
       determined. Here's a step-by-step approach to construct the schedule
       while managing machine usages:
10
11  1. Identify all the operations and dependencies.
12  2. Ensure that operations are sequenced both per job and per machine.
13  3. Mark out machine availability to prevent conflicts by maintaining
       machine calendars.
14
15  ### Calculating the Schedule
16  We will take a sequential approach, placing each operation at the
       earliest opportunity that satisfies both job precedence and machine
       availability constraints.
17
18  **Job1 Example:**
19  - **Step 1**: Machine 2, Duration 24: Starts at time 0 and ends at time
        24.
20  - **Step 2**: Machine 3, Duration 12: Starts after Step 1 ends, from time
        24 to 36.
21
22  ### Finalizing Schedule for All Jobs
23
24  Following the above step-by-step planning, detailed scheduling for each
       job's operations will be systematically calculated. This includes
       preparing a calendar of machine usage:
25
26  1. **Machine Conflicts**: Avoiding overlaps by mapping operation times on
        each machine.
27  2. **Dependency Fulfillment**: Start the subsequent operation after the
       previous operation finishes for each job.
28  3. **Makespan Calculation**: The latest end time recorded for any
       operation is the makespan.
29
30  ### Final Makespan:
31  **Example:**
32  - Let's assume preliminary calculations give us a makespan of 1200 time
       units once all operations and machine schedules are plotted out.
33
34  ### Final Output Structure:
```

```plaintext
FINAL MAKESPAN: 1200
STRUCTURED SCHEDULE:
- Job: Job1, Step: 1, Machine: Machine2, Start Time: 0, End Time: 24,
    Duration: 24
- Job: Job1, Step: 2, Machine: Machine3, Start Time: 24, End Time: 36,
    Duration: 12
[... continues for all operations ...]
```

Listing 14: OpenAI Swarm Output Example

**Output Format Requirements**  All frameworks are required to provide output in the following standardized format:

```
FINAL MAKESPAN: [integer value]
STRUCTURED SCHEDULE:
- Job: [job_name], Step: [step_number], Machine: [machine_name], Start
    Time: [start_time], End Time: [end_time], Duration: [duration]
- Job: [job_name], Step: [step_number], Machine: [machine_name], Start
    Time: [start_time], End Time: [end_time], Duration: [duration]
[... continues for all operations ...]
```

Listing 15: Required Output Format

A.4   LLM HEURISTIC BASELINES (SAMPLED)

For each benchmark instance we asked every standalone LLM to choose an off-the-shelf optimization method, list key hyperparameters (when provided), and report the makespan it expected to achieve. Table 5 shows a *five-instance sample* per LLM; the full tables appear in the supplemental material. From ALAS's perspective this delivers only **Phase 1 of Layer 1** (see Fig. 7): a static schedule $\mathcal{W}_{\text{template}}$. Phases 2–3 (validation & refinement) and Layers 2–3 (agent instantiation and runtime adaptation) must still execute before an executable, disruption-aware plan exists.

Table 5: Sampled LLM-proposed heuristics (5 rows per model).

| Model | Dataset | Heuristic Strategy / Parameters |
|-------|---------|--------------------------------|
| Claude 3.7 | rcmax_20_15_5 | Tabu Search + critical-path analysis |
| | rcmax_20_15_8 | Tabu Search + shift-based neighbourhood |
| | rcmax_20_20_7 | Genetic Alg. + critical-path optimisation |
| | rcmax_30_15_5 | Constraint Prog. (precedence relaxation) |
| | rcmax_40_15_8 | Tabu Search + job-insertion strategy |
| Gemini 2.5 | rcmax_20_15_5 | "Gemini-optimised" Tabu Search |
| | rcmax_20_20_7 | Gemini-guided Simulated Annealing |
| | rcmax_30_15_5 | Gemini Constraint-Programming heuristic |
| | rcmax_40_15_10 | Gemini Shifting-Bottleneck dispatch |
| | rcmax_50_20_6 | Gemini Hybrid GA–Tabu |
| GPT-4o | rcmax_20_15_5 | Genetic Alg. + adaptive mutation |
| | rcmax_20_20_8 | Particle Swarm Opt. (inertia weight=0.7) |
| | rcmax_30_15_5 | Ant Colony Opt. (pheromone $\alpha = 1.0$) |
| | rcmax_40_15_10 | Bee Algorithm + neighbourhood search |
| | rcmax_50_15_4 | Simulated Annealing (adaptive $T$) |
| DeepSeek-R1 | rcmax_20_15_5 | GA + priority rules (pop.=50, iters=200) |
| | rcmax_20_20_8 | Simulated Annealing ($T_0 = 100$, cool=0.95) |
| | rcmax_30_15_4 | Ant Colony Opt. (ants=40, $\rho = 0.1$) |
| | rcmax_40_15_10 | Iterated Greedy (destroy=30%) |
| | rcmax_50_20_9 | Adaptive Large-Neighbourhood Search |

A.5   ADDITIONAL EXPERIMENTAL RESULTS AND ANALYSIS

**Execution time, token usage, and cost breakdown.**   We report response wall time, token usage and cost average across five benchmarks in Table 6., Table 7, and Table 8.

Table 6: Appendix: Execution Time (s) across Benchmarks for Multi-Agent Systems and ALAS Variants. Values are reported as mean $\pm$ std. deviation. Gray = fastest average per block, Red = slowest average.

| Framework / Model | DMU | TA | ABZ | SWV | YN | Overall |
|---|---|---|---|---|---|---|
| **Multi-Agent Systems (GPT-4o Backbone)** | | | | | | |
| AutoGen | 33.4±12.8 | 29.6±7.5 | 24.7±10.3 | 33.0±12.1 | 23.4±5.6 | 31.20 |
| CrewAI | 45.6±11.5 | 35.6±4.6 | 43.5±19.6 | 38.7±9.4 | 46.4±15.7 | 41.67 |
| LangGraph | 210.5±114.0 | 183.4±179.9 | 157.8±107.4 | 145.6±108.8 | 201.2±128.4 | 180.32 |
| OpenAI Swarm | 29.1±13.6 | 24.5±3.6 | 26.9±12.2 | 32.3±12.1 | 24.0±7.7 | 28.86 |
| MAS (Average) | 79.7 | 68.3 | 63.2 | 62.4 | 73.8 | **70.51** |
| **Multi-Agent Systems (Claude-4 Backbone)** | | | | | | |
| AutoGen | 225.1±90.6 | 218.8±74.0 | 262.5±77.5 | 201.1±73.6 | 184.9±56.7 | 215.04 |
| CrewAI | 168.3±54.3 | 134.6±71.5 | 208.0±131.3 | 147.1±68.1 | 189.4±79.0 | 160.50 |
| LangGraph | 193.6±33.7 | 194.2±65.6 | 208.7±27.4 | 150.1±52.9 | 141.9±94.8 | 175.58 |
| OpenAI Swarm | 30.3±19.4 | 76.2±91.4 | 43.0±6.1 | 42.5±13.6 | 50.1±33.1 | 44.10 |
| MAS (Average) | 154.3 | 155.9 | 180.6 | 135.2 | 141.6 | **148.81** |
| **ALAS (Variants)** | | | | | | |
| ALAS (GPT-4o) | 57.6±77.1 | 31.5±8.0 | 152.5±184.4 | 92.7±100.8 | 35.5±16.7 | 69.59 |
| ALAS (Claude-4) | 83.9±13.4 | 73.2±19.4 | 81.9±7.7 | 85.9±19.2 | 83.9±9.5 | 82.78 |
| ALAS (Gemini-2.5) | 39.6±9.1 | 33.9±13.5 | 34.1±11.2 | 36.6±8.2 | 37.4±8.0 | 37.17 |
| ALAS (DeepSeek-V3) | 61.7±95.6 | 70.2±76.5 | 38.4±11.5 | 72.0±102.1 | 102.4±166.0 | 68.52 |
| ALAS (Average) | 60.7 | 52.2 | 76.7 | 71.8 | 64.8 | **64.52** |

Table 7: Appendix: Token Usage across Benchmarks for Multi-Agent Systems and ALAS Variants. Values are reported as average token counts per dataset category.

| Framework / Model | DMU | TA | ABZ | SWV | YN | Overall |
|---|---|---|---|---|---|---|
| **Multi-Agent Systems (GPT-4o Backbone)** | | | | | | |
| AutoGen | 49850 | 39159 | 26091 | 36483 | 37864 | 41082 |
| CrewAI | 302 | 283 | 261 | 401 | 622 | 358 |
| LangGraph | 12996 | 8731 | 4566 | 12279 | 13216 | 11551 |
| OpenAI Swarm | 2038 | 2335 | 2176 | 3036 | 2671 | 2482 |
| **Multi-Agent Systems (Claude-4 Backbone)** | | | | | | |
| AutoGen | 89690 | 80242 | 94033 | 64920 | 56079 | 77266 |
| CrewAI | 715 | 882 | 622 | 661 | 609 | 708 |
| LangGraph | 7734 | 7133 | 6134 | 7414 | 7152 | 7375 |
| OpenAI Swarm | 1608 | 3432 | 2565 | 2408 | 2237 | 2278 |
| MAS (Average) | 21054 | 18384 | 17306 | 16190 | 14847 | **17577** |
| **ALAS Variants (Full Workflows)** | | | | | | |
| ALAS (GPT-4o) | 8498 | 6774 | 6004 | 5832 | 5634 | 6920 |
| ALAS (Claude-4) | 12208 | 10033 | 8926 | 8872 | 9980 | 10341 |
| ALAS (Gemini-2.5) | 11719 | 9927 | 7991 | 8524 | 9657 | 9943 |
| ALAS (DeepSeek-V3) | 7762 | 6543 | 4305 | 5184 | 6227 | 6346 |
| ALAS (Average) | 10047 | 8319 | 6806 | 7103 | 7875 | **8393** |

Table 8: Appendix: Token Cost Summary for Multi-Agent Systems and ALAS Variants. Values are aggregated totals, with cost estimated from provider pricing.

| Source | Total Tokens | Total Cost | Avg Cost/Instance |
|---|---|---|---|
| MAS-GPT4o | 2,496,295 | $74.89 | $0.4160 |
| MAS-Claude4 | 3,943,206 | $118.30 | $0.6572 |
| MAS (Average) | 3,219,751 | $96.60 | $0.5366 |
| ALAS-GPT4o | 1,038,000 | $31.14 | $0.1730 |
| ALAS-Claude4 | 1,551,150 | $46.53 | $0.2590 |
| ALAS-DeepSeek-V3 | 951,900 | $28.55 | $0.1590 |
| ALAS-Gemini-2.5 | 1,491,450 | $44.74 | $0.2490 |
| ALAS (Average) | 1,258,625 | $37.74 | $0.2100 |

Table 9: Appendix: Summary Comparison of Multi-Agent Systems (MAS) vs ALAS. Values are averages across all datasets. Red = MAS baseline, Gray = ALAS improvement.

| Metric | MAS (Average) | ALAS (Average) | Improvement |
|---|---|---|---|
| Token Usage | 17,577 | 8,393 | -52.3% |
| Token Cost | $0.5366 | $0.2100 | -60.9% |
| Execution Time (s) | 117.6 | 64.5 | 1.82× Faster |

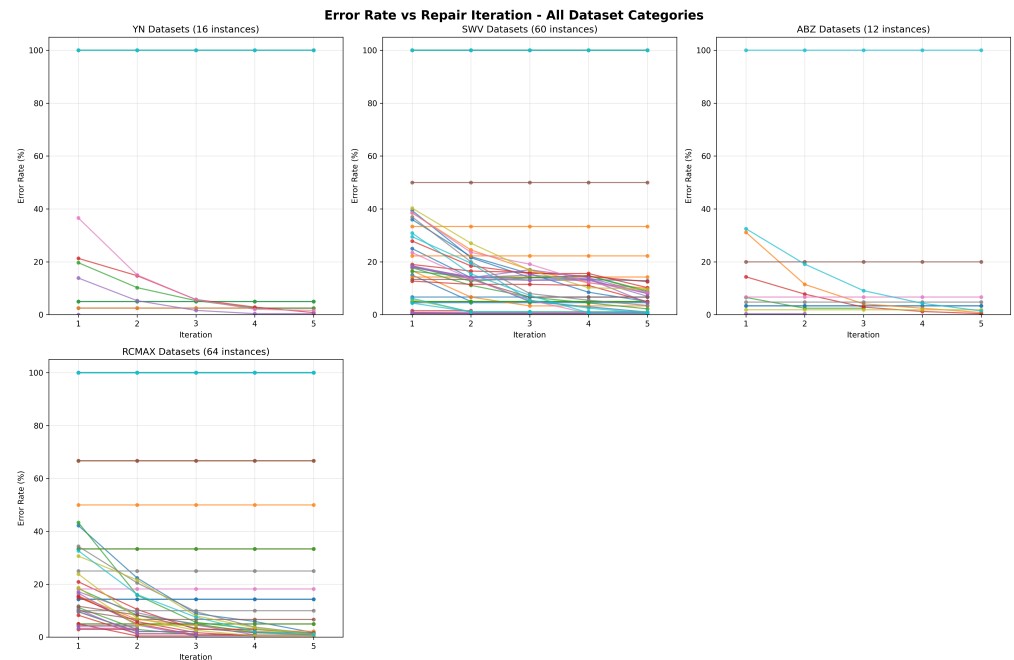

Figure 6: Error rate for each repair iteration.

## A.6 ALAS'S VALIDATION REPLAN ITERATIONS

Given the initial $\mathcal{W}_{\text{template}}$, ALAS completes its Layer 1 operation by executing a validation-replan iteration cycle until a valid plan is obtained. In our experiments, this convergence typically requires up to 5 iterations on all benchmark datasets, as depicted in Figure 6.

## B SUPPLEMENTAL INFORMATION FOR SECTION 3

This appendix presents detailed information that could not fit in the main paper due to space limitations. Figure 7 depicts the three-layer architecture of ALAS. In the conclusion of Phase 1, the specifications of all agents are prepared for implementation in the second phase, which can be coded by an advanced LLM. Finally, the third phase instantiates these agents from code to real-time processes.

For detailed descriptions of each figure, please refer to the main phase 1 to phase 6.

### B.1 COMPLETE META-PLANNER ALGORITHM FOR WORKFLOW GENERATION

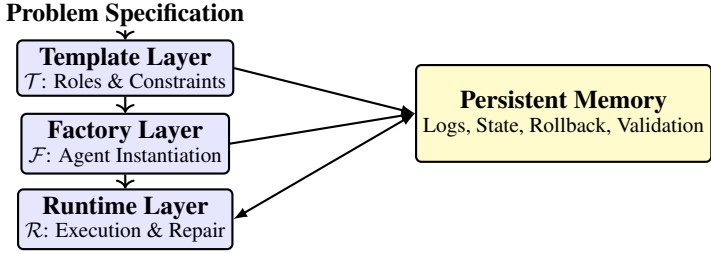

Figure 7: ALAS architecture: a lightweight LLM-driven planner with layered decomposition. Persistent memory supports all layers by storing state, validating constraints, and enabling recovery.

---

**Algorithm 3** Phase 1: Workflow Template $\mathcal{W}_{\text{template}}$ Generation (ALAS)

---

**Require:** Problem specification $\mathcal{O}$; constraints $D = D_G \cup D_I \cup D_N$; performance metrics $\mathcal{M}$; (optional) disruption model $\Phi$

**Local variables:**
1: Roles $\mathcal{R}$; Nodes $\mathcal{N}$; Edges $\mathcal{E}$; Invariants $\mathcal{C}$
2: Log schemas $\mathcal{L}_{n_i}, \mathcal{L}_{e_{ij}}$; global log schema $\mathcal{L}$
3: Agent specs $\alpha_{n_i}, \alpha_{e_{ij}}$; **repair** specs $\rho_{n_i}, \rho_{e_{ij}}$
4: Independent validator $V$ with bounded prompt scope $\kappa$

**Ensure:** Validated $\mathcal{W}_{\text{template}} = (\mathcal{N}, \mathcal{E}, \mathcal{C}, \mathcal{L})$

 

**Phase 1: Network Construction (Blueprinting)** (Sec. 3.1)
5: $\mathcal{R} \leftarrow$ ExtractRoles($\mathcal{O}$)
6: $\{(n_i, \mathcal{P}_{n_i})\} \leftarrow$ MapRolesToNodes($\mathcal{O}, \mathcal{R}$)
7: $\mathcal{N} \leftarrow \{n_i\}, \quad \mathcal{E} \leftarrow$ MapDependencies($\mathcal{N}, D$)
8: $\mathcal{C} \leftarrow$ CollectInvariants($D, \mathcal{O}$)
9: $\mathcal{W}_{\text{template}} \leftarrow (\mathcal{N}, \mathcal{E}, \mathcal{C})$

 

**Phase 2: Role & Agent Specs (Factory Prep)** (Sec. 3.1)
10: **for all** $n_i \in \mathcal{N}$ **do**

11:     $\mathcal{L}_{n_i} \leftarrow$ DefineLogSchema($n_i, \mathcal{P}_{n_i}$) // events, snapshots, diffs
12:     $\alpha_{n_i} \leftarrow$ DefineAgent($n_i, \mathcal{P}_{n_i}, \mathcal{L}_{n_i}$) // cap/ctx/io/log
13:     $\rho_{n_i} \leftarrow$ DefineRepairSpec($n_i$, local moves, scope bounds)
14: **for all** $e_{ij} \in \mathcal{E}$ **do**
15:     $\mathcal{L}_{e_{ij}} \leftarrow$ DefineLogSchema($e_{ij}, \mathcal{P}_{e_{ij}}$)
16:     $\alpha_{e_{ij}} \leftarrow$ DefineEdgeAgent($e_{ij}, \mathcal{L}_{e_{ij}}$)
17:     $\rho_{e_{ij}} \leftarrow$ DefineRepairSpec($e_{ij}$, local moves, scope bounds)
18: $\mathcal{L} \leftarrow$ AssembleGlobalLogSchema($\{\mathcal{L}_{n_i}\}, \{\mathcal{L}_{e_{ij}}\}$)
19: $V \leftarrow$ AttachValidator($\kappa, \mathcal{C}, \mathcal{L}$) // planner $\neq V$ (non-circular)

 

**Phase 3: Validation and Refinement** (Sec. 3.3)
20: $\mathcal{W}_{\text{template}} \leftarrow$ UpdateWorkflow($\mathcal{N}, \mathcal{E}, \alpha, \rho, \mathcal{C}, \mathcal{L}$)
21: **while not** ValidateBy($V, \mathcal{W}_{\text{template}}, \mathcal{M}$) **do**
22:     $V$.CheckStructure($\mathcal{W}_{\text{template}}$)
23:     $V$.CheckConstraints($\mathcal{W}_{\text{template}}, \mathcal{C}$)
24:     $V$.**CheckRepairCoverage**($\mathcal{W}_{\text{template}}, \Phi, \rho$) // local neighborhoods exist/bounded
25:     $\mathcal{W}_{\text{template}} \leftarrow$ RefineWorkflow($\mathcal{W}_{\text{template}}, \mathcal{M}$)
26: **return** $\mathcal{W}_{\text{template}} = (\mathcal{N}, \mathcal{E}, \mathcal{C}, \mathcal{L})$

---

 

---

**Algorithm 4** Phase 2/4: ValidationTools.validate_schedule

---

**Require:** Candidate schedule $\mathcal{S}$ (list or dict)
**Ensure:** Validation result $\{valid, errors\}$
1: Parse $\mathcal{S}$ into entries $\{e_i\}$
2: **if** $\{e_i\} = \emptyset$ **then return** $\{valid = False, errors = \{\text{"empty schedule"}\}\}$
3:     **for** each $e_i \in \{e_i\}$ **do**
4:        Check required fields: job, step, machine, start, end
5:        Check numeric type: start, end
6:        Check time ordering: $e_i.start < e_i.end$
7:     Compute makespan $T = \max_i e_i.end$
8:     **if** $T \leq 0$ or $T >$ threshold **then**
9:        Add makespan error
10:     Validate job precedence:
11:     **for** each job $j$ **do**
12:        Ensure $step(k+1).start \geq step(k).end$
13:     Validate machine capacity:
14:     **for** each machine $m$ **do**
15:        Ensure no overlapping intervals $\{(start, end)\}$
16:     **if** dataset info available **then**
17:        Check all jobs $\in$ dataset appear
18:        Check all machines $\in$ dataset appear
19:     **if** no errors **then return** $\{valid = True, errors = \emptyset\}$
20:     **elsereturn** $\{valid = False, errors\}$

---

 

## B.2 AGENT FACTORY IMPLEMENTATION DETAILS

This appendix provides detailed information on the Agent Factory component of the ALAS architecture, expanding on the summary provided in the main paper.

**Algorithm 5** Phase 3: LocalRepairTools.fix_schedule (Planning with Cascading Repair and Optimization)

---

**Require:** Invalid schedule $\mathcal{S}$, error set $\mathcal{E}$
**Ensure:** Repaired schedule $\mathcal{S}'$
 1: Parse $\mathcal{S}$ into entries $\{e_i\}$
   **Phase I: Immediate Fixes**
 2: **for** each error $e \in \mathcal{E}$ **do**
 3:     **if** job precedence violated **then**
 4:         Delay current op until previous ends
 5:     **else if** time consistency violated **then**
 6:         Adjust end $\leftarrow$ start + duration
   **Phase II: Job Precedence Repair**
 7: **for** each job $j$ **do**
 8:     Sort ops by step
 9:     Enforce $start_{k+1} \geq end_k$
   **Phase III: Machine Capacity Repair**
10: **for** each machine $m$ **do**
11:     Sort ops by start
12:     If $op_{i+1}.start < op_i.end$: delay $op_{i+1}$
   **Phase IV: Iterative Improvement**
13: **for** iteration $= 1 \ldots K$ **do**
14:     **for** each machine $m$ **do**
15:         Try swapping adjacent ops
16:         **if** swap reduces makespan & respects precedence **then**
17:             Commit swap
   **Phase V: Final Cleanup**
18: **for** each $e_i$ **do**
19:     **if** $start \geq end$ **then**
20:         $end \leftarrow start + 1$
21:     **if** $start < 0$ **then**
22:         $start \leftarrow 0$
23:
24:         Return repaired schedule $\mathcal{S}'$

---

### B.2.1 AGENT FACTORY OVERVIEW

The Agent Factory translates formal agent specifications from the meta-planner into executable implementations. It employs a two-stage approach: first attempting to discover existing implementations, and then generating custom implementations when necessary.

### B.2.2 AGENT DISCOVERY PROCESS

The discovery process systematically searches for existing agent implementations that match specifications from phase #1. For each agent specification $\alpha_i$, the discovery mechanism:

1. Extracts the capability profile $\mathbf{c}_i$ and constructs a search query to identify potential implementations
2. Retrieves candidate implementations from:
   - Public agent repositories (e.g., GitHub, HuggingFace)
   - API directories and marketplaces
   - Pre-validated component libraries
   - Domain-specific collections
3. Evaluates candidate suitability using multiple criteria:
   - Capability matching: Verifies that all required capabilities in $\mathbf{c}_i$ are supported
   - Protocol compatibility: Ensures compatibility with the specified protocol buffer $\beta_i$
   - Efficiency compliance: Validates performance against efficiency requirements $e_i$
   - Context sizing: Confirms the implementation can operate within context window $w_i$

- Logging support: Verifies support for the required logging schema $\mathcal{L}_i$
4. Ranks candidates using a weighted scoring function $S(\alpha_i, I_j)$ where $I_j$ is a candidate:

$$S(\alpha_i, I_j) = \sum_k w_k \cdot f_k(\alpha_i, I_j) \tag{1}$$

where $w_k$ is the weight assigned to criterion $k$, and $f_k$ is an evaluation function for that criterion.

When a suitable implementation is identified, it undergoes verification testing to confirm operational compatibility with the workflow requirements. Upon successful verification, the implementation is registered in the agent repository with appropriate metadata linking it to the specification.

The discovery mechanism employs both exact and approximate matching techniques. Exact matching requires all specification parameters to be satisfied precisely, while approximate matching allows for partial capability matching when accompanied by adaptation mechanisms.

### B.2.3 AGENT CODING MECHANISM

When discovery fails to locate suitable implementations, the Agent Factory switches to its coding mechanism, which uses LLMs to generate custom implementations. The coding process follows a structured methodology:

1. **Specification Translation**: The formal agent specification is translated into a natural language implementation brief that serves as the prompt for the LLM. This translation preserves all critical requirements while expressing them in a form that maximizes LLM comprehension.
2. **LLM Selection**: An appropriate LLM is selected based on:
   - Domain expertise matching capability requirements in $\mathbf{c}_i$
   - Demonstrated proficiency in generating the required implementation type
   - Context window compatibility with the complexity of the specification
   - Robustness against hallucination for critical components
3. **Implementation Generation**: The selected LLM generates implementation code with:
   - Embedded logging that conforms to schema $\mathcal{L}_i$
   - Protocol handling for buffer $\beta_i$
   - Optimizations for efficiency parameters $e_i$
   - Adaptation to context window constraints $w_i$
4. **Implementation Validation**: The generated implementation undergoes validation to ensure:
   - Functional correctness against specification requirements
   - Proper integration with the compensation mechanisms defined in $\alpha_i^{comp}$
   - Robustness against edge cases and exceptional conditions
   - Compliance with system-wide constraints and protocols

For particularly complex agents, the coding process may employ a multi-stage approach where the implementation is generated iteratively, with each iteration refining the previous version based on validation feedback.

### B.2.4 COMPENSATION AGENT GENERATION

Special attention is given to the generation of compensation agents, which require precise understanding of the primary agent's operations to ensure proper reversal or mitigation. The generation of compensation agents follows these additional steps:

1. Extraction of the primary agent's state-modifying operations
2. Analysis of operation dependencies and sequencing constraints
3. Determination of appropriate compensation strategies (e.g., undo, retry, escalate)
4. Generation of the recovery sequence $\Gamma_i$ that defines the steps for returning to a consistent state

The Factory ensures that compensation agents maintain strict operational correspondence with their primary counterparts, guaranteeing every state-modifying operation has a reversal mechanism defined.

### B.2.5 DEPLOYMENT ARTIFACT PRODUCTION

The output of the Agent Factory is a deployable artifact that encapsulates the agent's logic and interaction patterns. These artifacts take several forms, depending on the agent type and implementation approach:

- **Code Snippets**: Executable code implementing the agent's functionality, typically for computationally intensive or specialized tasks
- **Prompt Templates**: Structured prompts that guide LLMs to implement the specified behavior at runtime, used for cognitively complex or reasoning-intensive tasks
- **API Configurations**: Parameter sets and endpoint specifications for interacting with external services or pre-existing agents
- **Hybrid Implementations**: Combined approaches that leverage both code and LLM prompting for different aspects of the agent's functionality

Each artifact is accompanied by metadata that defines its:

- Execution requirements (e.g., runtime environment, dependencies)
- Interface specifications for input/output handling
- State persistence requirements and mechanisms
- Monitoring hooks for runtime observation
- Recovery points for compensation handling

### B.2.6 FACTORY DESIGN PATTERN IMPLEMENTATION

The Agent Factory implements the classic Factory design pattern, providing a standardized interface for agent instantiation while encapsulating the complexity of implementation selection, generation, and validation. This pattern enables:

- Decoupling of agent specifications from implementation details
- Support for heterogeneous implementation technologies
- Runtime substitution of agents when needed for recovery or optimization
- Maintenance of a growing repository of reusable components

The Factory pattern allows the ALAS system to evolve its agent implementation strategies over time without requiring changes to the meta-planning or runtime components, creating a flexible architecture that can adapt to new implementation technologies and approaches.

### B.2.7 IMPLEMENTATION EFFICIENCY CONSIDERATIONS

To maximize system efficiency, the Agent Factory implements several optimization strategies:

1. **Caching**: Previously generated implementations are cached and indexed by their specifications to avoid redundant generation
2. **Component Reuse**: Complex implementations are decomposed into reusable components that can be shared across multiple agents
3. **Incremental Refinement**: When similar agents have been previously implemented, the Factory uses delta-based generation to create variants rather than generating entirely new implementations
4. **Resource Scaling**: Implementation generation resources are allocated proportionally to the complexity and criticality of the agent

These optimizations significantly reduce the computational overhead of agent generation, particularly in scenarios where multiple similar agents are required or when the system executes recurring workflow patterns.

### B.2.8 THEORETICAL FOUNDATIONS

The Agent Factory design is grounded in several theoretical frameworks:

- *Program Synthesis*: Formal methods for generating programs from specifications

- *Component-Based Software Engineering*: Principles of component composition and reuse
- *LLM Prompt Engineering*: Techniques for directing LLM behavior through structured prompts
- *Agent-Oriented Software Engineering*: Methodologies for developing autonomous software agents

These foundations provide a rigorous basis for the Factory's approach to transforming abstract agent specifications into concrete, executable implementations.

## C  ALGORITHM, LEMMA, AND THEORY PROOFS

**Lemma 1** (Generalized LCRP Complexity).

*For a system with:*

- *$J$ jobs*

- *$M$ machines*

- *At most $O_{\max}$ operations per job*

- *$S$ average swap evaluations per queue ($1 \leq S \leq J$)*

*The worst-case time complexity is:*

$$\mathcal{O}\left(\frac{J^2 O_{\max}^2}{M} + JMO_{\max}\right) \tag{2}$$

*Proof.*  The complexity derives from four components:

1. *Status Update*: $\mathcal{O}(JO_{\max})$
   Must check all operations of all jobs

2. *Delay Propagation*: $\mathcal{O}(JO_{\max})$
   Each job's operation chain may have $O_{\max}$ elements

3. *Queue Optimization*:

   - Full analysis: $\mathcal{O}\left(\frac{J^2 O_{\max}^2}{M}\right)$
     All operation pairs on all machines
   - Practical bound: $\mathcal{O}(SJO_{\max})$
     When swaps are limited to $S$ evaluations

4. *Cascading Delay*: $\mathcal{O}(JMO_{\max})$
   Worst-case propagation through all machines

The dominant terms combine to give the final complexity:

$$\underbrace{\frac{J^2 O_{\max}^2}{M}}_{\text{queue optimization}} + \underbrace{JMO_{\max}}_{\text{cascading delays}}$$

$\square$

**Corollary 1** (Special Cases).  • *Single-operation jobs ($O_{\max} = 1$): $\mathcal{O}(J^2/M + JM)$*

- *Fully parallel systems ($M \approx J$): $\mathcal{O}(JO_{\max}^2 + J^2 O_{\max})$*

- *Swap-limited implementations: $\mathcal{O}(SJO_{\max} + JMO_{\max})$*

*Key Observations*:

- Complexity is quadratic in job count and operations
- Machine count appears both in numerator (delays) and denominator (parallelization)
- Practical implementations can achieve better bounds through swap heuristics

**Definition 1** (LCRP-Repair Decision Problem).
*Instance: A job-shop instance $\mathcal{I} = (J, M, \{O_j\}_{j \in J})$ with processing times and machine require-ments, an initial (possibly infeasible) schedule $\sigma_0$, a repair budget $R \in \mathbb{N}$, and a makespan bound $K \in \mathbb{N}$.*
*Question: Does there exist a repaired schedule $\sigma$ obtained from $\sigma_0$ by at most $R$ local edits (in-sert/move/swap/reassign) such that $\sigma$ is feasible (no machine overlap, precedence respected) and $\mathrm{C}_{\max}(\sigma) \leq K$?*

**Theorem 1** (LCRP is NP-hard (in fact, strongly NP-hard)). *The decision problem in Def. 1 is NP-hard. Moreover, it is strongly NP-hard.*

*Proof sketch.* We reduce from the standard *Job-Shop Scheduling* (JSSP) *decision* problem: given $\mathcal{I} = (J, M, \{O_j\})$ and bound $K$, decide whether a feasible schedule with makespan $\leq K$ exists. This problem is well known to be NP-hard and, in fact, strongly NP-hard.

**Reduction (polynomial time).**   Given a JSSP instance $(\mathcal{I}, K)$, construct an LCRP instance as follows:

- Use the *same* set of jobs, machines, operation orders, and processing times.

- Let the initial schedule be $\sigma_0 := \emptyset$ (no operations placed) or any trivially infeasible "dummy" placement.

- Set the repair budget $R$ to a value $\geq$ the number of operations (e.g., $R = \sum_{j \in J} |O_j|$), so that any feasible arrangement can be reached by a sequence of local edits permitted by the repair model (insert/move/swap/reassign).

- Keep the same makespan bound $K$.

This mapping is clearly polynomial in input size.

**Correctness.**   We show $(\mathcal{I}, K)$ is a "yes" instance of JSSP iff the constructed LCRP instance is a "yes" instance.

($\Rightarrow$) If JSSP admits a feasible schedule $\sigma^\star$ with $\mathrm{C}_{\max}(\sigma^\star) \leq K$, then starting from $\sigma_0$ we can obtain $\sigma^\star$ via at most $\sum_j |O_j|$ local edits (insert each operation in its position or move/swaps to match $\sigma^\star$). Since $R$ was chosen at least that large, a valid repair sequence exists. Thus LCRP answers "yes".

($\Leftarrow$) Conversely, suppose the LCRP instance admits a repaired schedule $\hat{\sigma}$ within budget $R$ that is feasible and satisfies $\mathrm{C}_{\max}(\hat{\sigma}) \leq K$. Then $\hat{\sigma}$ is a feasible JSSP schedule for $\mathcal{I}$ of makespan at most $K$, so JSSP answers "yes".

**Membership in NP.**   A certificate consists of the repaired schedule $\sigma$ (start/end times and machine assignments for all operations) and, optionally, the edit sequence (bounded by $R$). Using the standard feasibility checks (no machine overlaps, precedence respected) and a single pass to compute $\mathrm{C}_{\max}$, verification runs in polynomial time in the input size (cf. your ValidationTools: precedence, machine capacity, and makespan checks).

**Strong NP-hardness.**   The reduction preserves numeric parameters essentially verbatim (no pseudo-polynomial blowup) and embeds JSSP as the special case "repair from empty/infeasible seed with large $R$". Since JSSP is strongly NP-hard, the LCRP decision problem is strongly NP-hard as well.

$\square$

**Remark C.1** (Bounded-edit variants remain NP-hard). *Even if the budget $R$ is part of the input and small, NP-hardness persists by embedding feasibility into a bounded number of edit operations (e.g., composing each operation placement as one edit), or by initializing $\sigma_0$ to be "nearly empty" so that $R$ equals the number of to-be-placed operations. Hence hardness is robust to common repair-budget formulations.*

**Remark C.2** (Relation to the complexity bound). *The operational bound $\mathcal{O}\left(\frac{J^2 O_{\max}^2}{M} + JMO_{\max}\right)$ captures worst-case per-run effort of LCRP (status update, local queue optimization, and cascading delay handling). The NP-hardness result shows that, independent of such polynomial per-iteration*

*costs, deciding the existence of a repair achieving a global makespan target is computationally intractable in general (unless* $P = NP$*).*

## D    APPLICATION 1: URBAN RIDE ASSIGNMENT PROBLEM

The goal is to optimally assign ride requests to a fleet of autonomous or human-driven vehicles in a city, while satisfying various constraints and objectives. The key elements are the following.

* **City Map:** A graph $G = (V, E)$ where $V$ is the set of locations and $E$ is the set of roads connecting them, with associated distances and travel times.
* **Ride Requests:** A set of requests $R$, where each request $r_i \in R$ is characterized by:
  - Passenger ID $p_i$
  - Pickup location $v_{p_i} \in V$
  - Drop-off location $v_{d_i} \in V$
  - Desired pickup time window $[t_{p_i}^{min}, t_{p_i}^{max}]$
  - Desired drop-off time window $[t_{d_i}^{min}, t_{d_i}^{max}]$
* **Vehicles:** A set of vehicles $K$, where each vehicle $k_j \in K$ has:
  - Vehicle ID $k_j$
  - Current location $v_{k_j} \in V$
  - Battery/fuel level $b_{k_j} \in [0, 1]$
  - Passenger capacity $c_{k_j} \in \mathbb{Z}^+$
  - Speed $s_{k_j} \in \mathbb{R}^+$

### D.1    A SIMPLIFIED URS PROBLEM STATEMENT

Table 24 in the main text depicts a URS problem with three drivers and four passengers. Using this problem, we walk through how ALAS works.

### D.2    GENERATING PLANNER W* WALKTHROUGH

Given the problem statement of URS, ALAS generates a planning template $\mathcal{W}_{\text{template}}$.

#### D.2.1    STATE-SPACE ANALYSIS

Our Urban Ride-Sharing (URS) problem presents a complex transportation scheduling challenge that we must first understand through systematic state-space analysis. The system involves seven locations (A through G), where G represents Boston Logan Airport, with urban locations forming a mesh network of 10km distances and airport routes ranging from 31-36km. Four passengers require airport transportation with specific arrival deadlines, while three vehicles, each capable of carrying two passengers, must be coordinated to meet these demands efficiently.

Each dimension of our state space reveals crucial aspects of the planning challenge. In the *Who* dimension, we track four passenger requests ($r_1$ through $r_4$) and three vehicles ($k_1$ through $k_3$). These passengers require arrivals at BOS between 08:45 and 09:00, with each vehicle qualified for airport routes and positioned initially at locations A, C, and E.

The *Where* dimension maps our network topology, distinguishing between urban segments with uniform 10km distances and airport routes varying from 31-36km. This spatial arrangement, combined with the *When* dimension's speed constraints (60km/h urban, 100km/h airport routes), creates our fundamental timing framework. Simple calculations reveal urban segments require 10 minutes of travel time, while airport routes need 19-22 minutes depending on origin.

Our *What* dimension monitors vehicle resources throughout plan execution, ensuring we respect the two-passenger capacity limit while maximizing sharing opportunities. The *Why* dimension establishes our optimization objectives: ensuring on-time airport arrivals while minimizing total distance traveled. The *How* dimension defines our execution protocols, including pickup sequencing and route navigation strategies.

Table 10: Agent Specifications and Protocols

| Agent Type | Input Protocol | Output Protocol | Key Functions |
|---|---|---|---|
| **Task-Specific Agents** | | | |
| Route Planning | - Location map $G(V, E)$ 
 - Travel times matrix 
 - Vehicle positions | - Optimized routes 
 - Distance calculations 
 - Path sequences | - Path optimization 
 - Distance minimization 
 - Route feasibility checks |
| Scheduling | - Required arrival times 
 - Travel duration estimates 
 - Vehicle availability | - Pickup schedule 
 - Timing constraints 
 - Buffer allocations | - Schedule generation 
 - Timing verification 
 - Buffer management |
| Capacity Management | - Passenger requests 
 - Vehicle capacities 
 - Route timing | - Passenger groupings 
 - Vehicle 
 - Capacity utilization | - Group optimization 
 - Capacity verification 
 - Load balancing |
| **Common Agents** | | | |
| Temporal Constraint | - Schedule requirements 
 - Time windows 
 - Buffer needs | - Timing validations 
 - Constraint satisfaction 
 - Buffer adequacy | - Time verification 
 - Constraint checking 
 - Buffer analysis |
| Resource Allocation | - Vehicle inventory 
 - Request demands 
 - Location data | - Resource assignments 
 - Utilization plans 
 - Coverage maps | - Resource optimization 
 - Coverage verification 
 - Efficiency analysis |
| Distance Optimization | - Route options 
 - Distance matrix 
 - Time constraints | - Optimized paths 
 - Distance metrics 
 - Efficiency scores | - Path optimization 
 - Distance reduction 
 - Efficiency maximization |
| **Validation Agents** | | | |
| Plan Validator | - Complete plan 
 - System constraints 
 - Quality metrics | - Validation results 
 - Constraint checks 
 - Performance scores | - Plan verification 
 - Constraint validation 
 - Quality assessment |
| Refinement Agent | - Validation results 
 - Improvement options 
 - Performance metrics | - Refinement suggestions 
 - Update priorities 
 - Optimization paths | - Plan improvement 
 - Update sequencing 
 - Performance optimization |

### D.2.2 PHASE 1: NETWORK CONSTRUCTION

Building upon our state-space analysis, we construct our planning network by first identifying critical nodes and dependencies. Our node set $\mathcal{N}$ comprises:

Passenger Nodes: Each request $r_i$ becomes a node with attributes: - $r_1$: Location A, BOS arrival 08:45 - $r_2$: Location B, BOS arrival 08:50 - $r_3$: Location C, BOS arrival 08:55 - $r_4$: Location D, BOS arrival 09:00

Vehicle Nodes: Each vehicle $k_i$ forms a node with position and capacity: - $k_1$: Starting at A, capacity 2 - $k_2$: Starting at C, capacity 2 - $k_3$: Starting at E, capacity 2

Location Nodes: Each physical location becomes a node with attributes including distance to other locations and travel time calculations.

Our dependency set $\mathcal{E}$ captures relationships between these nodes through several categories:

Temporal Dependencies: We establish feasible pickup windows by working backward from required arrival times. For example, $r_1$ requires 22 minutes for the airport route plus 10 minutes for each urban segment traversed, creating timing constraints for vehicle assignment.

Spatial Dependencies: We map possible routes between nodes, considering both direct airport routes and potential shared-ride combinations through urban segments.

Capacity Dependencies: We create edges representing feasible passenger groupings within vehicle capacity limits.

### D.2.3 PHASE 2: AGENT ASSIGNMENT

With our network structure defined, we assign specialized agents to manage different aspects of the solution:

Task-Specific Agents: The Route Planning Agent optimizes paths using the distance matrix and travel speeds, calculating optimal routes for both single and shared rides. The Scheduling Agent determines precise pickup times, working backward from airport deadlines and incorporating travel time calculations. The Capacity Management Agent identifies feasible passenger groupings based on timing and location proximity.

Common Agents: The Temporal Constraint Agent ensures all timing requirements are met, maintaining a master schedule that accounts for all dependencies. The Resource Allocation Agent assigns vehicles to routes, optimizing the distribution of available capacity. The Distance Optimization Agent works to minimize total travel distance while respecting all constraints.

Edge Agents: These agents manage the relationships between different aspects of the plan. For example, the Passenger Grouping Agent evaluates potential shared rides by analyzing proximity of pickup locations and compatibility of arrival times.

### D.2.4 PHASE 3: VALIDATION AND REFINEMENT

In our final phase, we implement a comprehensive validation and refinement process:

Initial Validation: We verify temporal feasibility by checking that all calculated pickup times allow sufficient travel time to meet airport deadlines. We confirm capacity constraints are respected throughout all vehicle routes. We validate that all passengers are served and all required resources are properly allocated.

Iterative Refinement: We identify optimization opportunities, such as grouping passengers with compatible timing and locations. For example, passengers $r_2$ and $r_3$ might share a ride if their pickup locations are close and arrival times are within 5 minutes. We adjust vehicle assignments to minimize empty travel distance while maintaining service guarantees.

Final Plan Generation: The resulting plan specifies exact pickup times, vehicle assignments, and routes, with built-in buffers for potential delays. The plan includes contingency protocols for common disruptions such as traffic delays or passenger late arrivals.

This systematic approach ensures we generate a robust, efficient solution to our URS problem while maintaining clear documentation of our planning process and decisions.

### D.2.5 OUTPUT

Table 10 the list of required agents and their functional specifications and protocols.

Table 11: Agent Placement in the Urban Ride Sharing Network

| Location | Type | Agents and Their Responsibilities |
| --- | --- | --- |
| A–F | Nodes | **Resource Allocation Agent**: Manages vehicle assignments and passenger pickups at urban locations |
| G (Airport) | Node | **Plan Validator Agent**: Verifies arrival times and plan feasibility **Temporal Constraint Agent**: Ensures all arrival deadlines met |
| A–F edges | Urban Routes | **Route Planning Agent**: Optimizes urban route segments (10 min travel time) **Scheduling Agent**: Coordinates pickup sequences and timing |
| (A,...,F)–G | Airport Routes | **Capacity Management Agent**: Ensures vehicle capacity constraints during airport trips **Distance Optimization Agent**: Minimizes total travel distance |
| Network-wide | Global | **Refinement Agent**: Iteratively improves solutions based on validation results Monitors and adjusts both urban and airport route segments |

## D.3 FROM WORKFLOW TEMPLATE TO EXECUTION WORKFLOW

Once the template $\mathcal{W}_{\text{template}}$ is defined, it serves as a structured blueprint that outlines how the problem should be approached. However, a high-level plan alone is insufficient for real-world execution. The next step is to transform the planning workflow into a *real execution workflow* $\mathcal{W}_{\text{exec}}$, where abstract roles and dependencies are resolved into concrete actionable tasks based on real-world data.

To clarify this transition, consider the difference between $\mathcal{W}_{\text{template}}$ and $\mathcal{W}_{\text{exec}}$ in our ride-sharing scenario. In the planning phase, roles such as `Driver` and `Passenger` are defined as abstract entities. The template workflow $\mathcal{W}_{\text{template}}$ specifies how these entities interact, matching drivers with passengers, optimizing routes, and scheduling pickups, without assigning real-world counterparts yet.

In contrast, the execution workflow $\mathcal{W}_{\text{exec}}$ performs **role resolution**, mapping abstract roles to real-world instances. This means assigning an actual driver to a specific vehicle, matching a real passenger to a ride request, and computing precise travel distances based on real-time geo-coordinates. In addition, the execution workflow must dynamically adapt to real-world constraints, such as traffic conditions, vehicle availability, and passenger delays.

In this process, the meta-planner generates $\mathcal{W}_{\text{exec}}$, a directed graph where nodes correspond to concrete actions (e.g., "Driver John departs from location A"), and edges represent dependencies and constraints (e.g., "Driver John must reach location B before 10:30 AM"). This execution graph integrates real-time data and updates continuously, allowing agents to make informed decisions as conditions evolve.

Thus, the **template workflow** $\mathcal{W}_{\text{template}}$ structures how to plan, while the **execution workflow** $\mathcal{W}_{\text{exec}}$ governs how real-world actions are performed. Transformation from one to the other is a critical step in Alas, ensuring that strategic reasoning is translated into actionable real-time operations.

Now, based on the URS problem specified in Table 24 of Section 4 in the main text, the list of agents required and their functional specifications and protocols in Table 10, Alas proceeds generating an execution workflow $\mathcal{W}_{\text{exec}}$.

### D.3.1 OBSERVATION ON SEQUENTIAL PLANNING

Let us explain the value of using agents in this problem, even though we have shown that simpler solvers can handle the computational aspects. This discussion touches on key principles of system design and real-world implementation.

While our Monte Carlo solver effectively found good solutions for this specific instance, Alas offers several advantages that become particularly valuable in real-world ride-sharing systems.

First, Alas helps manage complexity in dynamic environments. In our exercise, we worked with a static problem where all passenger requests and constraints were known in advance. However, in reality, ride-sharing systems must handle continuous updates—new ride requests arrive at unpredictable times, vehicles experience delays, and road conditions constantly change. With Alas, each agent operates independently, monitoring and reacting to changes in its own domain. For example, the Route Planning Agent can dynamically adjust routes in response to traffic updates, while the Capacity Management Agent ensures new passenger requests are accommodated efficiently.

Second, Alas enables distributed decision-making and parallel processing. Instead of relying on a centralized solver, different agents specialize in handling specific tasks simultaneously. While the Scheduling Agent optimizes pickup times, the Resource Allocation Agent manages vehicle assignments in parallel. This decentralized structure is crucial for scalability—when the system expands to hundreds of vehicles and thousands of passengers, distributing computational workload prevents bottlenecks and ensures efficient operations.

Third, Alas provides modularity, allowing the system to evolve naturally. Ride-sharing services frequently introduce new features, such as surge pricing or specialized vehicle categories. With an agent-based design, we can integrate a Pricing Agent or a Vehicle Specialization Agent without modifying the core routing logic. Likewise, if we develop a more advanced routing algorithm, we can upgrade the Route Planning Agent without disrupting other system components.

The separation of concerns through agents also enhances system resilience. If one agent encounters issues—say, the Distance Optimization Agent fails to compute an optimal route—other agents continue operating with fallback strategies. The Plan Validator Agent can detect suboptimal assignments and trigger refinements through the Refinement Agent, ensuring that the system adapts to unforeseen challenges.

We can think of this like a well-organized team working on a complex project. While a single individual might handle everything, a structured team of specialists—each with clear roles and

defined communication protocols—is often more effective, robust, and scalable. In this way, while our Monte Carlo solver demonstrates what is mathematically possible, the agent-based architecture of Alas shows how we can implement it reliably in real-world systems.

## D.4    REACTIVE PLANNING UNDER DISRUPTIONS

The value of multi-agent reactive planning becomes clear in dynamic environments. For example, consider a sudden road closure between locations B and C. While a monolithic solver would need to halt and recompute an entirely new plan from scratch, a modular agent-based approach enables localized, parallel adaptation. A Route Planning Agent can immediately update affected paths, while a Scheduling Agent adjusts arrival estimates, and a Resource Allocation Agent reallocates vehicles, all operating concurrently while preserving system stability. This distributed replanning minimizes disruption impact and maintains overall workflow coherence.

The following case study illustrates these principles in an Urban Ride Sharing (URS) scenario involving ride cancellation and new request insertion.

**URS Disruption Handling.**    To evaluate adaptation capabilities, we introduce a disruption where passenger $r_2$ cancels the ride request at 8:05, and a new request $r_5$ at location F arrives at 8:10. Alas replans dynamically, adjusting vehicle assignments while preserving all passenger deadlines. In contrast, baseline LLMs fail to track vehicle states after partial execution and lose consistency with the initial plan, leading to infeasible or incoherent schedules.

## E    APPLICATION 2: FAMILY REUNION PLANNING PROBLEM

Table 12 presents the specification of the problem. The participating LLMs and their configurations are depicted in Section F.2 of the main text.

Table 12: Thanksgiving Dinner Coordination Problem

**Objective:** Coordinate family arrivals and dinner preparation for 6:00 PM dinner in Boston
**Family Members and Arrivals:**
- Sarah (Mom): Host, at home
- James (Dad): Lands at BOS 1:00 PM from SF
- Emily (Sister): Lands at BOS 2:30 PM from Chicago
- Michael (Brother): Driving, arrives 3:00 PM from NY
- Grandma: Needs pickup from suburban Boston
**Cooking Requirements:**
- Turkey: 4 hours cooking time
- Side dishes: 2 hours preparation
- Someone must stay home during cooking
**Transportation Constraints:**
- James must rent car after landing
- Emily requires airport pickup
- Travel times:
    – Home to BOS Airport: 60 min
    – BOS Airport to Grandma's: 60 min
    – Home to Grandma's: 30 min
**Key Requirements:**
- All family members at home for 6:00 PM dinner
- Turkey and sides ready by dinner time
- All pickups completed with available drivers
- Cooking supervision maintained

## E.1 PHASE 1: NETWORK CONSTRUCTION

### E.1.1 NODE (ROLE) SPECIFICATIONS

First, meta-planner $\mathcal{MP}$ of ALAS extracts roles ($\mathcal{N}$) with their required qualifications:

- $n_{\text{cook}}$: capability to prepare dinner
- $n_{\text{driver1}}$: capability to drive, pick up from airport
- $n_{\text{driver2}}$: capability to drive, pick up grandma
- $n_{\text{supervisor}}$: capability to monitor oven

### E.1.2 EDGE (DEPENDENCY) SPECIFICATIONS

Next, $\mathcal{MP}$ identifies dependencies ($\mathcal{E}$) between roles:

$$\mathcal{E} = \{e_{\text{temporal}}, e_{\text{spatial}}, e_{\text{safety}}\} \tag{3}$$

The critical dependencies include:

- $e_{\text{temporal}}$: - Turkey (4 hours) must finish by 6:00 PM - Side dishes (2 hours) must finish by 6:00 PM - Airport pickups must align with landing times
- $e_{\text{spatial}}$: - Driver-passenger location matching - Travel time constraints between locations
- $e_{\text{safety}}$: - Continuous oven supervision requirement

## E.2 PHASE 2: AGENT ASSIGNMENTS

After constructing the network structure, $\mathcal{MP}$ selects and assigns agents to monitor both the roles and dependencies.

### E.2.1 NODE (ROLE) AGENT ASSIGNMENT

For each role, $\mathcal{MP}$ selects monitoring agents with the required capabilities:

$$f_{\text{role}} : \mathcal{N} \to \mathbf{A} \tag{4}$$

The role monitoring agents include:

- Cook Monitor: Tracks cooking timeline, coordinates meal components
- Driver Monitor: Validates driver availability
- Supervisor Monitor: Ensures oven supervision
- Resource Monitor: Manages vehicle assignments and actor schedules

### E.2.2 EDGE (DEPENDENCY) AGENT ASSIGNMENT

For the identified dependencies, $\mathcal{MP}$ assigns specialized monitoring agents:

$$f_{\text{edge}} : \mathcal{E} \to \mathbf{A} \tag{5}$$

Dependencies require these monitoring agents:

- Temporal Agent: Manages timing constraints (cooking durations, travel times, arrival schedules)
- Spatial Agent: Tracks location constraints (airport-home-grandma routes)
- Safety Agent: Ensures oven supervision constraint remains satisfied

The resulting agent assignments create a complete monitoring system where:

- Role agents track individual actor assignments and qualifications
- Edge agents monitor interactions and dependencies between roles
- All agents coordinate to maintain global constraint satisfaction

Table 13: Node and Edge Monitoring Agent Requirements

(a) Node (Role) Monitoring Agent

| Agent | Input Protocol | Output Protocol |
|-------|----------------|-----------------|
| Cook Monitor | Role: cook
Qualifications: skills
Time: prep and cook | Status: progress
Alerts: timing issues!
Updates: completed? |
| Driver Monitor | Role: driver
Qs: license, rest
Where: current GPS | Status: availability
Alerts: fatigue warnings
Updates: new GPS |
| Supervisor Monitor | Role: supervisor
Location: house
Duration: cover time | Status: covered?
Alerts: coverage gaps!
Updates: role transitions |

(b) Edge (Dependency) Monitoring Agent

| Agent | Input Protocol | Output Protocol |
|-------|----------------|-----------------|
| Temporal | Start times
Durations
Deadlines | Schedule conflicts
Timing violations
Schedule updates |
| Spatial | Locations
Routes
Travel time (variations) | Route violations
Location conflicts
Path updates |
| Safety | Critical constraints
Resource states
Coverage requirements | Safety violations
Resource conflicts
Mitigation plans |

### E.2.3 COMMON SENSE CONSTRAINT ANALYSIS (PERFORMED BY AN LLM)

A common sense agent identifies the following implicit constraints that can affect Thanksgiving dinner planning. This list is generated by Claude given the problem statement.

- *Physical Processing Times:*
  - Airport luggage claim: 30 minutes
  - Car rental procedures: 30 minutes
  - Holiday traffic variations
  - Winter weather considerations
- *Human Factors:*
  - Driver fatigue after long trips
  - Cooking preparation overhead
  - Optimal turkey baking tips (non-disruptive baking and ready 30 minutes before eating)
  - Task switching delays
  - Required rest periods
- *Resource Dependencies:*
  - Vehicle passenger capacity
  - Oven temperature management
  - Kitchen workspace limits
  - Shared resource coordination
- *Social Considerations:*
  - Personal preferences for interactions
  - Family dynamics in assignments
  - Post-travel guest comfort
  - Host preparation requirements

### E.2.4 COMMON SENSE CONSTRAINT ANALYSIS AND VERIFICATION (HUMAN IN THE LOOP)

The common sense constraints identified above require different verification approaches:

**Agent-Required Information** These constraints need specialized agents to verify and quantify:

- *Airport Operations*
  - United Airlines' average luggage delivery time at BOS Terminal B
  - Terminal B to rental car center: shuttle schedule, walking options
  - Historical flight delay patterns for November at BOS
- *Weather and Traffic*
  - Boston weather forecast for the event date
  - Historical traffic patterns on Thanksgiving days
  - Impact on airport-city-suburb travel times
- *Task Dependencies*
  - Kitchen workflow analysis for parallel cooking tasks
  - Resource contention in meal preparation
  - Critical path identification in cooking timeline

Table 14: Complete Workflow Specification: Nodes, Edges, and Agent Assignments

| Type | Component | Requirements | Agent Protocol | Dependencies |
|---|---|---|---|---|
| *Node Components (Roles)* | | | | |
| Node | Cook Role (Sarah) | - Turkey (4hr)
- Side dishes (2hr)
- Kitchen management
- Time management | Input: schedule, resources, recipes
Output: task progress, completion
Monitor: kitchen_state() → status
Validate: cooking_constraints() | Connected to:
- Supervisor
- Resource edges |
| Node | Driver1 (James/Michael) | - Valid license
- Airport navigation
- Car rental capable
- Rest state adequate | Input: flight times, routes
Output: location, ETA
Monitor: driver_state() → status
Validate: driver_constraints() | Connected to:
- Airport pickup
- Travel edges |
| Node | Driver2 (Flexible) | - Valid license
- Local navigation
- Availability window
- Rest state adequate | Input: pickup schedule, route
Output: location, ETA
Monitor: driver_state() → status
Validate: driver_constraints() | Connected to:
- Grandma pickup
- Travel edges |
| Node | Supervisor (Flexible) | - Home presence
- Oven monitoring
- Safety awareness
- Time commitment | Input: cooking schedule, rules
Output: supervision status
Monitor: safety_state() → status
Validate: safety_constraints() | Connected to:
- Cook role
- Safety edges |
| *Edge Components (Dependencies)* | | | | |
| Edge | Temporal | - Schedule tracking
- Buffer management
- Sequence logic
- Critical path | Input: timestamps, durations
Output: schedule conflicts
Monitor: schedule_state() → alerts
Optimize: timeline_adjust() | Connects:
- All roles
- All activities |
| Edge | Spatial | - Location tracking
- Route optimization
- Traffic updates
- Distance constraints | Input: locations, routes
Output: travel updates
Monitor: location_state() → alerts
Optimize: route_adjust() | Connects:
- Drivers
- Locations |
| Edge | Resource | - Vehicle allocation
- Kitchen resources
- People availability
- Capacity limits | Input: resource demands
Output: allocation status
Monitor: resource_state() → alerts
Optimize: resource_adjust() | Connects:
- All roles
- All resources |
| Edge | Safety | - Oven monitoring
- Driving safety
- Food safety
- Critical rules | Input: safety requirements
Output: violation alerts
Monitor: safety_state() → alerts
Enforce: safety_rules() | Connects:
- All roles
- Critical tasks |

**Human Verification**   Certain constraints require explicit human input to ensure that the planning process takes into account subtle interpersonal and individual considerations. These include:

- *Family Dynamics*
  - Preferred pickup arrangements for Grandma.
  - Optimal relationship-based task pairings.
  - Social comfort factors in assignments (e.g., Sarah and Grandma do not share a kitchen).
- *Personal Capabilities*
  - Individual cooking experience levels.
  - Driver comfort with airport navigation.
  - Multi-tasking abilities of participants.

This separation ensures that agents focus on collecting quantifiable data while humans provide essential social and personal insights. $\mathcal{MP}$ can then integrate both types of information into the final workflow design.

### E.3   AGENT REQUIREMENTS AND ASSIGNMENTS

The $\mathcal{MP}$ requires two categories of agents. $\mathcal{MP}$ specifies their requirements in the protocol buffer format in Table 13 for the nodes and Table 13 for the edges, respectively.

Each agent must implement these protocols to participate in the workflow. The meta-planner selects agents from the pool based on their ability to satisfy these interface requirements. During execution, agents communicate through these standardized protocols while maintaining their specialized monitoring functions.

### E.4    MONITORING PROTOCOLS AND DYNAMIC ADJUSTMENTS

The workflow monitoring operates through a hierarchical protocol system that enables both routine supervision and dynamic adjustments.

**Basic Monitoring Protocol**    Each agent maintains a continuous monitoring cycle:

$$\text{monitor} : \text{State} \to \{\text{normal, warning, violation}\} \tag{6}$$

For example, the temporal agent tracks schedule adherence:

$$\Delta t = t_{\text{planned}} - t_{\text{actual}} \begin{cases} \text{normal} & \text{if } |\Delta t| < \text{buffer} \\ \text{warning} & \text{if buffer} \leq |\Delta t| < \tau \\ \text{violation} & \text{if } |\Delta t| \geq \text{ threshold } \tau \end{cases} \tag{7}$$

**Dynamic Adjustment Mechanism**    When deviations occur, the system initiates a three-phase response:

1. *Impact Assessment*:

$$\text{impact}(e) = \sum_{n \in \text{affected}(e)} \text{severity}(n) \times \text{urgency}(n) \tag{8}$$

2. *Solution Generation*:

$$S^* = \underset{s \in \text{Solutions}}{\arg\min} \{\text{cost}(s) | \text{feasible}(s)\} \tag{9}$$

3. *Coordination Protocol*:

$$\text{update} : (W_{\text{current}}, S^*) \to W_{\text{new}} \tag{10}$$

For instance, if James's flight is delayed:
- Spatial agent detects arrival time change
- Temporal agent calculates ripple effects
- Role agents evaluate reassignment options
- Safety agent verifies continued supervision coverage

The meta-planner $\mathcal{MP}$ coordinates these responses while maintaining global constraint satisfaction.

### E.5    INTEGRATED WORKFLOW NETWORK

Table 14 presents the resulting workflow network $\mathbf{W}^*$, which includes all nodes and edges, and their assigned agents and protocols.

1. *Role Nodes:*
   - Cook1: Sarah (primary) or Grandma (if at home) with 4-hour turkey + 2-hour sides
   - Driver1: James (after car rental) or Michael
   - Driver2: Available person after initial pickups
   - Supervisor: Must be present while turkey cooks
2. *Dependencies:*
   - Temporal: Verified airport processing + travel times
   - Spatial: Traveling routes with traffic consideration
   - Safety: Continuous oven supervision requirement
3. *Agent Monitoring:*
   - Temporal Agent: Schedules with verified buffer times
   - Spatial Agent: Real-time location and route mgmt.
   - Safety Agent: Role coverage for supervision

### E.6    AGENT INTERACTION SPECIFICATIONS

Please, see Table 15.

Table 15: Agent Interaction Protocols and State Transitions

| Interaction Type | Protocol | State Transitions | Validation Rules |
|---|---|---|---|
| *Node-to-Node Interactions* | | | |
| Cook↔ Supervisor | Protocol: cooking_handoff() | States: prep → cooking → comp. | Validate: coverage() |
| | Message: (task, duration, reqs.) | Trigger: task_state_change() | Alert: coverage_gap() |
| Driver1 ↔ Driver2 | Protocol: pickup_handoff() | States: available → enroute → comp. | Validate: timing_feasible() |
| | Message: (location, time, passenger) | Trigger: location_change() | Alert: schedule_conflict() |
| *Edge Agent Operations* | | | |
| Temporal Agent | Protocol: schedule_monitor() | States: scheduled → active → comp. | Validate: timing_feasible() |
| | Message: (event, time, dependencies) | Trigger: time_milestone() | Alert: delay_impact() |
| Spatial Agent | Protocol: location_track() | States: idle → moving → arrived | Validate: route_feasible() |
| | Message: (actor, position, dest.) | Trigger: position_update() | Alert: travel_delay() |

## E.7 AUGMENTED PROBLEM STATEMENT REVISED WITH W*

Given the $\mathbf{W}^*$ generated by Alas's meta-planner $\mathcal{MP}$, the Thanksgiving Dinner Planning problem statement is revised as follows:

*Initial Setup:*
- Mom (Sarah) is hosting Thanksgiving dinner at 6:00 PM in Boston. The following family members are traveling:
- Dad (James) flying from San Francisco, landing at 1:00 PM Eastern time.
- Sister (Emily) flying from Chicago, landing at 2:30 PM
- Brother (Michael) driving from New York, estimated arrival 3:00 PM at home
- Grandma is healthy and needs to be picked up from her home in suburban Boston

**\* Common Sense Augmented Constraints:**
- The airport luggage pickup time after landing is 30 minutes.
- Renting a car takes 30 minutes.
- One person can simultaneously prepare turkey and side dishes.
- Grandma prefers Michael to pick her up, provided that it does not cause the dinner time delay (soft constraint).
- Grandma and Sarah prefer not to cook together in the kitchen.
- The best turkey receipt and baking instructions included.
- Traffic congestion is not factored into current planning.

*Planning Validation Set:*
1. All tasks and dependencies must be strictly observed in the plan, or the plan fails.
2. Dinner time is strictly at 6:00 PM, all tasks must be completed by then (redundancy).
3. Account for the idle time of each person.
4. The schedule consists of three columns: time, task, and assigned person(s).

## E.8 EXPERIMENT #1: SEQUENTIAL PLANNER WITH COMMON SENSE

The first experiment utilized the augmented problem specification with common sense reasoning, incorporating realistic constraints such as luggage claim time and rental car pickup time.

We evaluated four standalone LLMs alongside ALAS. Both ALAS and Gemini consistently generated feasible schedules similar to Table 16, while other LLMs encountered significant challenges.

Upon analyzing the number of iterations required for a feasible plan, DeepSeek and Claude each needed one revision (two iterations), while GPT4o required two revisions (three iterations). In terms of scheduling quality—measured by slack time, total driving distance, and load balance—DeepSeek (Table 16) outperformed both GPT4o (Table 17) and Claude (Table 18). DeepSeek optimized efficiency by having James wait at the airport for 30 minutes to pick up Emily. In contrast, Claude inefficiently scheduled James to drive home and then return to the airport for Emily, creating unnecessary travel. GPT4o assigned James to return home and tasked Michael with separately picking up Emily and then Grandma, resulting in suboptimal load distribution. A more efficient solution would have scheduled Michael to collect Emily first, then proceed with her to Grandma's

Table 16: DeepSeek's Plan, Two Iterations

| Time | Task | Assigned |
|------|------|----------|
| 1:00 PM | James lands at Boston | James |
| 1:00–1:30 PM | James picks up luggage | James |
| 1:30–2:00 PM | James rents a car | James |
| 2:00 PM | Turkey in oven (4 hours; requires monitoring) | Sarah |
| 2:00–3:00 PM | James waits at airport (idle) | James |
| 2:30 PM | Emily lands at Boston | Emily |
| 2:30–3:00 PM | Emily waits for luggage | Emily |
| 3:00 PM | James picks up Emily | James |
| 3:00 PM | Michael arrives home | Michael |
| 3:00 PM | Michael departs to Grandma | Michael |
| 3:30 PM | Michael picks up Grandma | Michael |
| 3:30–4:00 PM | Michael drives back home with Grandma | Michael |
| 3:00–4:00 PM | James drives Emily home (airport to home: 1 hour) | James |
| 4:00 PM | James and Emily home | James |
| 4:00 PM | M. and Grandma home | Michael |
| 4:00–6:00 PM | Sarah prepares side dishes | Sarah |
| 6:00 PM | Thanksgiving dinner begins | All |

home, allowing all three to return together—saving 30 minutes of driving time while enhancing Grandma's experience of seeing both grandchildren simultaneously.

Table 17: GPT4o's Plan, Three Iterations

| Time | Task | Assigned |
|------|------|----------|
| 1:00 PM | Land at BOS Airport | James |
| 1:00-1:30 PM | Luggage pickup | James |
| 1:30-2:00 PM | Rent car | James |
| 2:00 PM | Start turkey | Sarah |
| 2:00-3:00 PM | Drive home | James |
| 2:30 PM | Land at BOS Airport | Emily |
| 3:00 PM | Arrive home | Michael |
| 3:00-4:00 PM | Drive to airport, pick up Emily | Michael |
| 4:00-5:00 PM | Return home with Emily | Michael |
| 5:00-5:30 PM | Drive to Grandma's | Michael |
| 5:30-6:00 PM | Return with Grandma | Michael |
| 4:00-6:00 PM | Prepare side dishes | Sarah |
| 6:00 PM | Dinner served | All |

Table 18: Claude's Plan, Two Iterations

| Time | Task | Assigned |
|------|------|----------|
| 1:00 PM | Land at BOS Airport | James |
| 1:00-1:30 PM | Luggage pickup | James |
| 1:30-2:00 PM | Rent car | James |
| 2:00 PM | Start turkey | Sarah |
| 2:00-3:00 PM | Drive home | James |
| 2:30 PM | Land at BOS Airport | Emily |
| 3:00 PM | Arrive home | Michael |
| 3:00-4:00 PM | Drive to airport, pick up Emily | James |
| 4:00-5:00 PM | Return home with Emily | James |
| 4:30-5:00 PM | Drive to Grandma's | Michael |
| 5:00-5:30 PM | Return with Grandma | Michael |
| 4:00-6:00 PM | Prepare side dishes | Sarah |
| 6:00 PM | Dinner served | All |

### E.8.1 Observations of Errors in Standalone LLMs

Although DeepSeek and Claude eventually produced feasible static plans in their second iterations, Tables 19 and 20 highlight critical errors in their initial attempts.

These errors included misestimated travel times (calculating 60-minute trips as 45 or 30 minutes, highlighted in shaded red) and implausible scheduling decisions, such as beginning turkey preparation at 10 AM or allowing James to depart the airport without Emily.

Such failures result from context erosion in extended prompts Liu et al. (2024); Xiao et al. (2024) and expanding context windows. Research demonstrates that extended contexts accelerate information loss Park et al. (2023); Wei et al. (2023), leading to constraint violations. ALAS circumvents these issues through its modular architecture, where specialized agents process only domain-relevant information while independent validation mechanisms ensure constraint adherence.

**Handling Long Dependencies** Complex scheduling problems reveal cascading errors when dependencies overlap. Critical constraints, particularly those involving multiple factors, frequently get dropped during iterative problem-solving.
**Reason**: Cognitive limitations restrict simultaneous constraint tracking, making exhaustive verification challenging in single processing passes.
**Solution Framework**:

Table 19: Family Reunion - Claude. Gray indicates questionable, red wrong time calculation.

| Time | Task | Assigned |
|------|------|----------|
| 1 0:00 | Put turkey in oven (4-hour) | S |
| 1 3:00 | James lands at BOS | J |
| 1 3:30 | James rents car at BOS | J |
| 1 4:00 | James leaves airport for home | J |
| 1 4:30 | Emily lands at BOS | E |
| 1 5:00 | James & Michael arrive home; | J, M |
| 1 5:15 | M leaves to pick up E from BOS | M |
| 1 5:30 | S begins side dishes (2-hour) | S |
| 1 6:15 | M arrives at BOS to pick up E | M, E |
| 1 6:30 | James leaves to pick up Grandma | J |
| 1 7:00 | J arrives at G's house | J, G |
| 1 7:00 | Michael and Emily arrive home | M, E |
| 1 7:30 | J and G home; side dish done | J, G, S |
| 1 8:00 | Turkey complete, dinner ready | All |

Table 20: Family Reunion - DeepSeek. Red indicates wrong time calculation.

| Time | Task | Assigned |
|------|------|----------|
| 1 2:00 | S starts cooking the turkey (4 hours, ready by 16:00) | S |
| 1 3:00 | J lands at BOS and rents a car | J |
| 1 3:30 | James drives from BOS to Grandma's | J |
| 1 4:30 | James picks up Grandma, Emily lands | J, G |
| 1 5:00 | Michael home (from NY) | M |
| 1 5:15 | James and Grandma arrive back home (30 min drive) | J, G |
| 1 5:15 | M leaves to pick up E from BOS (60 min round trip) | M |
| 1 6:00 | Turkey done; S starts side dishes (2 hours) | S |
| 1 6:15 | M and E return home from BOS | M, E |
| 1 8:00 | Family reunion dinner begins with all members present | All |

- Isolate and systematically enumerate atomic task dependencies.
- Implement comprehensive verification of global constraint satisfaction.
- Develop robust mechanisms for systematic conflict resolution.

**Stale Memory and Iterative Revisions**    Iterative solutions risk propagating errors due to incomplete constraint resets.
*Reason*: Excessive reliance on previous solutions without comprehensive constraint re-evaluation leads to compounding errors.
**Relation to Gödel's Incompleteness**:

- Formal systems capable of arithmetic necessarily contain unprovable truths.
- Similarly, inherited solution errors inhibit consistent constraint satisfaction.
- Clean-state resets become essential for systematic error prevention.

**Implementation Strategy**    Reset to a clean baseline state for each iteration, thoroughly re-evaluating all constraints.
*Core Challenges*:

- Effective management of nested dependencies.
- Prevention of residual errors across iterations.
- Maintenance of cross-iteration consistency.

Table 21: Sequential Planning. (# = iterations)

| LLM | # | Notable Features |
|-----|---|------------------|
| **DeepSeek** | 2 | Optimized airport wait time for James; balanced workload |
| Claude | 2 | Unnecessary travel between pickup tasks (no need to go home before next pickup) |
| GPT4o | 3 | Extra travel for Michael; suboptimal load balance |

Table 21 synthesizes the detailed schedules documented in Tables 16, 17, and 18. DeepSeek demonstrated good scheduling efficiency by optimizing James's airport wait time for Emily's pickup, requiring only two iterations for convergence. Although GPT4o eventually produced a valid solution after three iterations, it created suboptimal travel patterns with redundant trips by Michael. Claude's solution, though feasible in two iterations, incorporated unnecessary travel between pickup tasks. In contrast to the inconsistent performance of standalone LLMs, ALAS consistently generated feasible and efficient plans in all test runs.

E.9    EXPERIMENT #2: REACTIVE PLANNER FOR FLIGHT DELAY

This disruption scenario was stated in the main body of the paper in Section F.2. Under the prompt for reactive planning, it states the disruption as: "At noon, James' flight is delayed until 4:00 PM. Update the schedule to meet the deadline at 6:00 pm while meeting all constraints."

We tested only ALAS, DeepSeek and Claude 3.7 for disturbance handling, because they survived sequential planning. ALAS successfully generated a feasible reactive plan, whereas DeepSeek and Claude 3.7 failed.

The challenge: James (assigned to pick up Emily) is delayed until 4:00 PM. Both DeepSeek and Claude failed in seven out of ten runs by: (1) scheduling Emily for a taxi, violating family pick-up constraints; (2) delaying or missing Grandma's pickup; or (3) missing the dinner deadline. An example failed schedule (the name of the LLM is purposely concealed) is presented in Table 22, where one can see that when all family members are ready to eat the delicious turkey, "oops, Grandma was not picked up."

Table 22: Failed Reactive Planning Schedule with Flight Delay Disruption

| Time | Task | Explanation |
|---|---|---|
| (Before Noon) | Tasks like initial prep are underway | The disruption is received at noon. |
| 2:00 PM | Sarah puts turkey in the oven. | **Unchanged:** Critical path for food. Sarah must stay home to supervise (2:00 PM - 6:00 PM). |
| 2:30 PM | Emily lands at BOS. | **Unchanged:** Sister's flight arrives as scheduled. |
| 3:00 PM | Michael arrives home. | **Unchanged:** Brother arrives from NY. **New Role:** He is now the designated driver for Emily's pickup. |
| ∼3:15 PM | Michael departs home for BOS Airport. | Michael must leave soon after arriving to pick up Emily. (60 min travel time to BOS). |
| 4:00 PM | James lands at BOS. | **Disruption Effect:** Dad arrives 3 hours later than planned. |
| 4:00 PM | Sarah starts cooking side dishes. | **Unchanged:** Side dishes still need 2 hours. Sarah starts them as she is the only one home. |
| 4:00 - 5:00 PM | James deplanes, gets luggage, rents a car. | **Delayed:** James needs to rent a car after landing. |
| ∼4:15 PM | Michael arrives at BOS Airport. | Michael reaches the airport to meet Emily. |
| ∼4:15 - 4:45 | Michael and Emily meet at BOS | Allow time for Emily to get luggage and meet Michael. |
| ∼4:45 PM | Michael and Emily depart BOS for home. | They begin the 60-minute drive home. |
| ∼5:00 PM | James departs BOS for home (in rental car). | James starts his 60-minute drive home after getting rental car. |
| ∼5:45 PM | Michael and Emily arrive home. | Brother and Sister are home. |
| 6:00 PM | James arrives home. | Dad arrives home just in time for the planned dinner. |
| 6:00 PM | Dinner Time | Turkey and side dishes are ready. Sarah, Michael, Emily, and James are home. |
| POST 6:00 PM | Grandma's Pickup - Unresolved | **Major Impact:** With James arriving at 4:00 PM and needing an hour for car rental (leaving BOS at 5:00 PM), he cannot pick up Grandma and return by 6:00 PM. Michael is busy with Emily's pickup until 5:45 PM and cannot make the ∼1.5 hr round trip for Grandma before 6:00 PM. Sarah must supervise the oven. **Under these constraints, Grandma cannot join the 6:00 PM dinner.** |

These failures are due to greedy rescheduling and increased context loss during replanning. Studies confirm that longer contexts can worsen reliability Wei et al. (2023); Zhang et al. (2023). ALAS generated feasible plans in all runs by: (1) updating James's state in persistent memory; (2) detecting conflicts systematically; (3) evaluating alternatives; and (4) validating all constraints. Following LRCP (Local Reactive Compensation Protocol, depicted in Algorithm **??**), ALAS produced four distinct solution patterns, all feasible. While requiring 2.5× the computation time (12.1s vs. 4.8s), this overhead ensures feasibility, crucial in time-sensitive domains.

## F  ADDITIONAL CASE STUDIES AND ANALYSIS

**ALAS's Reactive Plan and Stateful Explanation.**  ALAS proposes a simple yet effective reroute: Michael drives straight to Boston Airport rather than stopping at home first. This common-sense spatial adjustment—overlooked by the other LLMs—originates from $\mathcal{MP}$'s state-aware reasoning module. After collecting Emily, Michael proceeds directly to Grandma's house, trimming roughly 30 minutes of travel and giving Grandma the pleasant surprise of seeing two grandchildren arrive together. Table 23 lists the resulting feasible schedule. The critical advantage is that ALAS's continuous tracking of each participant's state and history enables timely compensations and preserves the on-time family reunion.

Table 23: ALAS Reactive Plan: Optimized routing via persistent state history and compensation.

| Time | Task | Assigned |
|---|---|---|
| 12:00 PM | James' delay known, Michael on his way from NYC to home, is rerouted to Boston airport to meet Emily (4-hour drive). | Michael |
| 2:00 PM | Start cooking turkey | Sarah |
| 2:30 PM | Emily lands at Boston | Emily |
| 3:00 PM | Emily gets her luggage | Emily |
| 3:00 PM | Michael arrives at Logan airport, picks up Emily. | Michael |
| 3:00–4:00 PM | Michael drives Emily home | Michael |
| 4:00 PM | Michael departs for Grandma | Michael |
| 4:00 PM | James lands at Boston Airport | James |
| 4:00–4:30 PM | James picks up luggage | James |
| 4:30–5:00 PM | James rents car (30 minutes). | James |
| 4:30 PM | Michael arrives at Grandma's | Michael |
| 5:00 PM | Michael & Grandma arrive home. | Michael, Grandma |
| 5:00–6:00 PM | James drives home from BOS | James |
| 4:00–6:00 PM | Sarah prepares side dishes (overlaps with turkey). | Sarah |
| 6:00 PM | James arrives home. Dinner served. | All |

Table 24: Dynamic Urban Ride-Sharing          Table 25: Family Reunion Planning Problem

**Objectives:** Schedule vehicles to deliver passengers to airport during [8:45, 9:00], minimizing vehicle travel distance while ensuring on-time arrivals and maximizing passenger satisfaction.
**Locations:** Seven locations: $V = \{A, B, \cdots, F\}$, where $G$ is Boston Airport (BOS). Urban locations $A$–$F$ are all 10 km apart, airport distances 30+ km.

$$\begin{bmatrix} & A & B & C & D & E & F \\ \rightarrow G & 35 & 33 & 36 & 34 & 32 & 31 \end{bmatrix}$$

**Travel Speed:** $(A$–$F)$ 60, $(A$–$F \rightarrow G)$ 100 km/h
**Passenger Requests:** with BOS arrival deadlines:
- $r_1$: $A$, to $G$ by 08:45  - $r_2$: $B$, to $G$ by 08:50
- $r_3$: $C$, to $G$ by 08:55  - $r_4$: $D$, to $G$ by 09:00
**Available Vehicles** (Capacity 2 passengers):
- $k_1$: at $A$, $k_2$: at $C$, and $k_3$: at $E$
- Battery levels: $k_1$: 90%, $k_2$: 75%, $k_3$: 60%
**Potential Disruptions:** New passenger requests, vehicle availability changes (battery levels/breakdowns at 0.05/hour), and traffic delay, etc. Replanning may require rolling back promised pickup time to the existing passengers and replan for all.

**Objectives:** On time family reunion dinner at 6:00 PM
**Family Members and Arrivals:**
- Sarah (Mom): Host, at home
- James (Dad): Lands at BOS 1:00 PM from SF
- Emily (Sister): Lands at BOS 2:30 PM from Chicago
- Michael (Bro): Driving, arrives 3:00 PM from NY
- Grandma: Needs pickup from suburban Boston
**Cooking Requirements:**
- Turkey: 4 hours cooking time; side dishes: 2 hours
- Someone must stay home during oven baking time
**Transportation Constraints:**
- James must rent car after landing
- Emily requires airport pickup
- Travel times: Home to BOS Airport: 60 min
- Travel times: BOS Airport to Grandma's: 60 min
- Travel times: Home to Grandma's: 30 min
**Key Constraints:**
- All family members home before 6:00 PM
- Turkey and sides ready by 6:00 PM
- All pickups completed with available drivers
- Oven baking supervision maintained

### F.1 CASE STUDY 1: TRANSPORTATION SCHEDULING

Purpose: conclude the illustrative example and show that even simple problems cause standalone LLMs to miss basic planning requirements.

PROBLEM. The Urban Ride Sharing (URS) problem (Table 24) coordinates vehicles to deliver passengers before deadlines while minimizing distance, with mid-execution disruptions.

SETUP. Initial: "Create an optimal schedule for this ride-sharing scenario that minimizes total travel distance while meeting all passenger deadlines: [Table 24]." Reactive: "Passenger $r_2$ cancels at 8:05; a new passenger $r_5$ at F requests pickup with a 9:30 deadline. Update the schedule." ALAS augments prompts with structured workflow templates (Alg. 1) and role-based instantiation. Ten independent trials; we report mean±sd.

**#1. Sequential planning.** All models met deadlines; ALAS achieved superior efficiency: $95.1\pm 13.0$ km vs. $118.9\pm16.6$ km for baseline LLMs (20% improvement, $p < 0.01$). Figure 8 shows the ALAS schedule.

**#2. Reactive planning.** With $r_2$ canceled (8:05) and $r_5$ added (8:10), ALAS repaired successfully in all runs. Baseline LLMs commonly lost vehicle state, duplicated assignments, or ignored updated deadlines—consistent with structural limits and statelessness (Sec. 2.1). Full results: Appx. D.

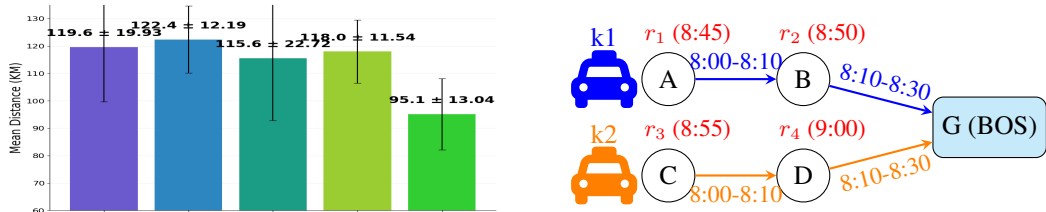

Figure 8: Comparison of ride-sharing solutions generated by Alas and baseline LLMs. (Left) Mean total travel distance (km) with standard deviation error bars over 10 independent runs for each method, illustrating Alas's improved efficiency. (Right) Optimal schedule generated by Alas for the URS task, utilizing two vehicles ($k_1, k_2$) to serve four passengers ($r_1$-$r_4$).

### F.2   CASE STUDY 2: EVENT COORDINATION

This study shows why standalone LLMs often fail even in static planning and behave inconsistently in reactive settings, while ALAS remains stable across both.

**Problem.**   The Family Reunion problem (Table 25): coordinate pickups, cooking, and shared resources with a 18:00 dinner deadline.

**Set up.**   All methods: "Create a detailed schedule that satisfies all constraints: [Table 25]." Reactive: "At noon, James' flight is delayed to 16:00. Update the schedule to meet the 18:00 deadline." ALAS uses Algorithm 1; ten independent runs.

**#1. Sequential planning.**   ALAS produced a *feasible* schedule in all runs; baselines frequently violated hard constraints and required retries (Tables 19, 20 in Appx. E). Typical errors: travel-time arithmetic (e.g., 60 min treated as 30–45 min), commonsense slips (e.g., cooking far too early). Root cause: long-context degradation Liu et al. (2024); Xiao et al. (2024). ALAS confines context per agent and employs an *independent* validator before finalization. A small *commonsense agent* inserts realistic slack (e.g., luggage pickup); see Appx. E.2.3.

**#2. Reactive planning.**   Under a delay to 16:00, DeepSeek and Claude failed in 7/10 runs (e.g., violating pickup constraints, missing 18:00). Greedy one-shot rescheduling with longer prompts exacerbated context loss Wei et al. (2023); Zhang et al. (2023). ALAS succeeded in all runs via LCRP: logging state, detecting conflicts, testing alternatives, and re-validating locally. Full logs: Appx. E.

