# OpenReview forum: "ALAS: Multi-Agent LLM Planning System via Validator Isolation and Localized Cascading Repair Protocol"
_ICLR.cc/2026/Conference — ICLR 2026 Conference Desk Rejected Submission_

### Official Review · Reviewer_yPcN · 2025-10-30

**Soundness:** 2
**Presentation:** 2
**Contribution:** 3
**Rating:** 4
**Confidence:** 2

**Summary:**

This paper presents ALAS, a multi-agent planning framework characterized by three key features:
(1) Validator isolation. Planning and verification are decoupled: an independent validator evaluates outputs to prevent self-approval and mitigate long-context degradation.
(2) Localized repair. When disruptions arise, a repair protocol computes and applies minimal, localized edits that preserve ongoing progress and task constraints.
(3) Versioned execution logs. The system maintains detailed records of state transitions and restore points, enabling transparent updates and targeted recovery without relying on long contexts.
Experiments on urban ride-sharing and job-shop scheduling demonstrate that ALAS achieves effective and efficient planning, maintaining robustness under disruptions.

**Strengths:**

1.	The motivation clearly identifies the limitations of LLM-based planning and naturally motivates the design principles behind ALAS.
2.	The ALAS framework is described thoroughly, with sufficient technical detail for understanding its mechanisms.
3.	The analysis of the LCRP mechanism is clear, showing its effectiveness and efficiency.

**Weaknesses:**

1.	Baseline methods:

(a)	LangGraph and CrewAI are treated as baseline methods, but they are frameworks for building multi-agent systems rather than specific methods. The paper does not provide details on how these baselines are instantiated for comparison.

(b)	The authors note that classical methods outperform LLMs on certain planning benchmarks but only include LLM-based baselines in their experiments.

2． Experimental results:

(a)	There is no definition for the optimal rate, making it unclear why this metric can exceed the success rate.

(b)	In Table 1, the notation for * is missing. Additionally, the success rate of ALAS(DeepSeek-V3) on ABZ equals that of DeepSeek-V3 (single agent) but is still labeled as significantly better than the baseline.

(c)	The success rate of ALAS varies substantially across different LLMs and benchmarks, yet no explanation or analysis is provided for these discrepancies.

2.	Ablation study: The ablation study includes only two simple variants. Additional ablations, such as circular validation and global repair, would help validate the contributions of ALAS’s individual components.

3.	Writing:

(a)	The beginning of Section 3 states that ALAS has a three-layer architecture, but five layers are subsequently described.

(b)	Citations in Section 4 are incorrectly formatted (e.g., “Demirkol–DMU Demirkol et al. (1998b;a)”), and there are abbreviations like  “LangGraph lan (2025).”

**Questions:**

1.	Can ALAS work on real-world planning tasks without disruption?

---

### Official Review · Reviewer_T92y · 2025-11-01

**Soundness:** 2
**Presentation:** 1
**Contribution:** 2
**Rating:** 2
**Confidence:** 3

**Summary:**

Existing LLM-based multi-agent planning struggles with reliability issues like verification, repair, and replanning. This paper introduces ALAS, a multi-agent LLM planning framework that allows a separation between validation and planning. To accomplish this, localized cascading repair with versioned execution logs is used to improve the reliability of the system. Additionally, the authors apply minimal local edits instead of using global replanning. This helps to avoid circular verification and reduce disruption costs. The authors run multiple experiments on urban ride-sharing and job-shop scheduling benchmarks. This exhibits great success, having lower token usage, faster execution, and strong robustness under dynamic changes compared to state-of-the-art Single-LLM and Multi-Agent System baselines

**Strengths:**

1.	The paper introduces validator isolation and localized cascading repair, addressing long-standing issues of circular verification and global recomputation in LLM planning, which may lead to unstable convergence, no trustworthy ground truth, and high computational cost.
2.	This paper shows that the system can perform robustly, even with disruption. For example, the localized repair protocol keeps the “blast radius” small. This helps to preserve work-in-progress, which leads to a graceful degradation, compared to previous models where the full system would collapse..
3.	Unlike previous black-box multi-agent frameworks, ALAS is unique. It is a white-box evaluation. This means it has insight into fine-grained visibility for faults, capabilities to edit neighborhoods, and the ability to repair steps. All of this makes debugging more transparent and less difficult.
4.	The authors evaluate the framework not just on toy domains but on realistic and challenging benchmarks like job-shop scheduling and urban ride-sharing. This provides more credibility towards the author's claims, demonstrating scalability and applicability in a wider market.

**Weaknesses:**

1. While the authors introduce a clean and modular architecture through ALAS. It remains unclear
why these specific mechanisms effectively address the stated challenges and how they ensure
robustness in practice. It would have been insightful for the authors to dive deeper into this and
provide more technical insight into their methods.

2. While the paper demonstrates strong performance improvements, it does not provide a deeper
analysis of what types of planning or reasoning failures ALAS can or cannot handle. It would have
been nice to have more analysis into typical planning or reasoning examples/cases.

3. Although the ablation study highlights the importance of the validation and repair modules, it does
not explore the underlying reasons behind the performance drop when either component is
removed. This limits the interpretability of the results, leaving some of it up to ambiguity, similar to
con 2

**Questions:**

1. The reliability study briefly mentions the concept of fault injection. However, it provides little qualitative
insight into why or how the validator succeeds or fails in these edge cases. A more detailed error
analysis, including the types of failures encountered and how the system recovers, would strengthen the
interpretability of the results.


2. Providing qualitative experiments and case studies would better demonstrate the superiority of ALAS
over baseline methods, helping readers understand not just that it works better, but how and why it does
so.

---

### Official Review · Reviewer_VbDJ · 2025-11-01

**Soundness:** 2
**Presentation:** 2
**Contribution:** 2
**Rating:** 2
**Confidence:** 2

**Summary:**

This paper introduces Alas, a multi-agent LLM planning framework to improve planning reliability. This is done through a 5-stage pipeline that (1) constructs a workflow template of how to perform the task (2) validates that the template is correct in a fresh context window (3) executes and locally repairs the plan during execution (4) revalidates the plan after each repair and (5) chooses the best schedule.

**Strengths:**

The paper describes the algorithm in great depth and psuedocode, provides experiments with baselines and ablations of the system components, and has an extensive appendix with case studies.

**Weaknesses:**

- This framework is very complex
- There are little to no gains over the single-agent baseline
  - The Best Variant per Dataset row is misleading; the equivalent Best Variant per Dataset for Single-Agent Models is 81.6% which corresponds to a 2.1% gain using ALAS
- The five main datasets tested on are all under the Job Shop Scheduling setting; it is unclear whether this framework would generalize to other domains

**Questions:**

My main concern is I don't think the gain in performance over the main benchmark justifies the complexity of this framework.
- Why do the corresponding Single-Agent models perform the same or better than ALAS in Table 1?
  -  Is Claude-Sonnet 4 and Alas(Claude-4) using the same model? If so, why is the performance drastically different here? The framework appears to be sensitive to the choice of model.
- How do Single-Agent models optimal rates look like in Table 2?

---

### Note · Program_Chairs · 2026-01-17
**Submission Desk Rejected by Program Chairs**

The following references in this submission do not refer to real documents and/or have major errors in bibliographic information:

 Siddharth Zhang, Daniel M. Do, and Dan Hendrycks. Be careful what you wish for: On the dangers of overoptimizing context length. arXiv preprint arXiv:2310.10631, 2023.